# Spatiotemporal *stop-and-go* dynamics of the mitochondrial TOM core complex correlates with channel activity

Shuo Wang[1], Lukas Findeisen [1], Sebastian Leptihn [2], Mark I. Wallace [3], Marcel Hörning [4✉] & Stephan Nussberger [1✉]

Single-molecule studies can reveal phenomena that remain hidden in ensemble measurements. Here we show the correlation between lateral protein diffusion and channel activity of the general protein import pore of mitochondria (TOM-CC) in membranes resting on ultrathin hydrogel films. Using electrode-free optical recordings of ion flux, we find that TOM-CC switches reversibly between three states of ion permeability associated with protein diffusion. While freely diffusing TOM-CC molecules are predominantly in a high permeability state, non-mobile molecules are mostly in an intermediate or low permeability state. We explain this behavior by the mechanical binding of the two protruding Tom22 subunits to the hydrogel and a concomitant combinatorial opening and closing of the two β-barrel pores of TOM-CC. TOM-CC could thus represent a β-barrel membrane protein complex to exhibit membrane state-dependent mechanosensitive properties, mediated by its two Tom22 subunits.

[1] Department of Biophysics, Institute of Biomaterials and Biomolecular Systems, University of Stuttgart, Stuttgart, Germany. [2] Zhejiang University-University of Edinburgh (ZJU-UoE) Institute, Zhejiang University, Zhejiang, China. [3] Department of Chemistry, King's College London, Britannia House, London, UK. [4] Department of Biobased Materials, Institute of Biomaterials and Biomolecular Systems, University of Stuttgart, Stuttgart, Germany. ✉email: Marcel.Hoerning@bio.uni-stuttgart.de; Stephan.Nussberger@bio.uni-stuttgart.de

The TOM complex of the outer membrane of mitochondria is the main entry gate for nuclear-encoded proteins from the cytosol into mitochondria[1]. It does not act as an independent entity, but in a network of interacting protein complexes, which transiently cluster in mitochondrial outer- and inner-membrane contact sites[2,3]. For proteins destined for integration into the lipid bilayer of the inner mitochondrial membrane, TOM transiently cooperates with components of the inner-membrane protein translocase TIM22. Proteins en route to the mitochondrial matrix require supercomplex formation with the inner-membrane protein translocase TIM23[4,5]. Depending on the activity of mitochondria, the lateral organization and the possibility of the respective repositioning of TOM, TIM22, and TIM23 in the outer and inner mitochondrial membranes to form transient contact sites may therefore be of fundamental importance for the different import requirements of the organelle[2].

Detailed insights into the molecular architecture of the TOM core complex (TOM-CC) from N. crassa[6], S. cerevisiae[7,8] and human[9] mitochondria were obtained by cryo-electron microscopy (cryoEM). All three structures show well-conserved symmetrical dimers, where the monomer comprises five membrane protein subunits. Each of the two transmembrane β-barrel domains of the protein-conducting subunit Tom40 interacts with one subunit of Tom5, Tom6, and Tom7, respectively. Two central transmembrane Tom22 receptor proteins, reaching out into the cytosol and the mitochondrial intermembrane space (IMS), connect the two Tom40 pores at the dimer interface.

Most studies that reported on the dynamic properties of the TOM-CC channel have been based on ion current measurements through single TOM-CC channels in planar lipid membranes under application of a membrane potential[10–14]. However, the physiological significance of the voltage-dependent conformational transitions between the open and closed states of the TOM-CC is controversial because the critical voltage $|\Delta V| > 50$ mV above which TOM-CC channels close[14,15], is significantly greater than any metabolically theory-derived potential $|MDP| < 5$ mV at the outer mitochondrial membrane[16].

In this work, we probe and correlate lateral mobility and ion flux though single TOM-CC molecules in lipid membranes resting on ultrathin hydrated agarose films using non-invasive real time electrode-free optical single-channel recording[17,18]. This approach does not require the use of fluorescent fusion proteins or labeling of TOM-CC by fluorescent dyes or proteins that might interfere with lateral movement and function of TOM-CC in the membrane. In our study, possible changes in ion flux associated with conformational changes of the TOM-CC are only caused by thermal motion of the protein and possible interactions with the hydrogel underlying the membrane. Our setup is therefore a simplified model system where we study a single aspect of complex interactions in the mitochondrial membrane.

We find that freely diffusing TOM-CC molecules stall when interacting with structures adjacent to the membrane, ostensibly due to interaction between the extended polar domains of Tom22 and the supporting agarose film. Concomitantly with suspension of movement, TOM-CC changes reversibly from an active (both pores open) to a weakly active (one pore open) and inactive (both pores closed) state. The strong temporal correlation between lateral mobility and ion permeability suggests that TOM-CC channel gating is highly sensitive to molecular confinement and the mode of lateral diffusion. Taken alongside recent cryo-electron microscopy of this complex, we argue that these dynamics provide a new functionality of the TOM-CC. From a general perspective, for the best of our knowledge, this may be the first demonstration of β-barrel protein mechanoregulation and the causal effect of lateral diffusion. The experimental approach we used can also be readily applied to other systems in which channel activity plays a role in addition to lateral membrane diffusion.

## Results

**Visualizing the open-closed channel activity of TOM-CC.** TOM-CC was isolated from a N. crassa strain that carries a version of subunit Tom22 with a hexahistidine tag at the C-terminus (Fig. 1a)[6,10,19], and reconstituted into a well-defined supported lipid membrane[20]. Droplet interface membranes (DIBs) were created through contact of lipid monolayer-coated aqueous droplets in a lipid/oil phase and a lipid monolayer on top of an agarose hydrogel (Supplementary Fig. S1)[17,18]. The cis side of the membrane contained $Ca^{2+}$-ions, while having a $Ca^{2+}$-sensitive fluorescent dye (Fluo-8) at the trans side. $Ca^{2+}$-ion flux through individual TOM-CCs was measured by monitoring Fluo-8 emission in close proximity to the membrane using TIRF microscopy in the absence of membrane potential to avoid voltage-dependent TOM-CC gating (Fig. 1b, c). Contrary to the classical single-molecule tracking approach using single fluorescently labeled proteins, an almost instantaneous update of fluorophores close to the TOM-CC nanopores can be observed. This enables the spatiotemporal tracking of individual molecules for an observation time up to several minutes.

Upon TIRF illumination of membranes with 488 nm laser light, single TOM-CC molecules appeared as high-contrast fluorescent spots on a dark background (Fig. 2a). High ($S_H$), intermediate ($S_I$), and low ($S_L$) intensity levels were indicating $Ca^{2+}$-flux through the TOM-CC in three distinct permeability states. No high-contrast fluorescent spots were observed in membranes without TOM-CC. The fact that the TOM-CC is a dimer with two identical β-barrel pores[6–9] suggests that the high and intermediate intensity levels correspond to two conformational states ($S_H$ and $S_I$) with two pores and one pore open, respectively. The low intensity level may represent a conformation ($S_L$) where both pores are closed with residual permeation for calcium.

Supplementary Movie S1 shows an optical recording of the open-closed channel activity of several TOM-CCs over time. Individual image frames of membranes were recorded at high frequency at a frame rate of $47.5 \, s^{-1}$ and corrected for fluorescence bleaching. The position and amplitude of individual spots were determined by fitting their intensity profiles to a two-dimensional symmetric Gaussian function with planar tilt to account for possible local illumination gradients in the bleaching corrected background (Fig. 2b, Supplementary Fig. S2). The time evolution of amplitude signals (Fig. 2c, d, Supplementary Movie S2 and Fig. S3) shows that the TOM-CC does not occupy only one of the permeability states $S_H$, $S_I$ and $S_L$, but can switch between these three permeability states over time.

To rule out the possibility that the observed intensity fluctuations are caused by possible thermodynamical membrane undulations in the evanescent TIRF illumination field or by local variations in $Ca^{2+}$ flux from cis to trans, we compared the ion flux through TOM-CC with that through isolated individual Tom40 molecules[21] and an unrelated multimeric β-barrel protein[22] which has three pores and is almost entirely embedded in the lipid bilayer. In a series of control experiments, we reconstituted Tom40 (Fig. 2e, f and Supplementary Fig. S4) and E. coli OmpF (Fig. 2e, f and Supplementary Figs. S5–S7) into DIB membranes and observed virtually constant fluorescence intensities, respectively. In contrast to TOM-CC, neither protein channel exhibits gating transitions between specific permeability states. The toggling of the TOM-CC between the three different permeability states $S_H$, $S_I$ and $S_L$ (Fig. 2c) therefore must have another root cause.

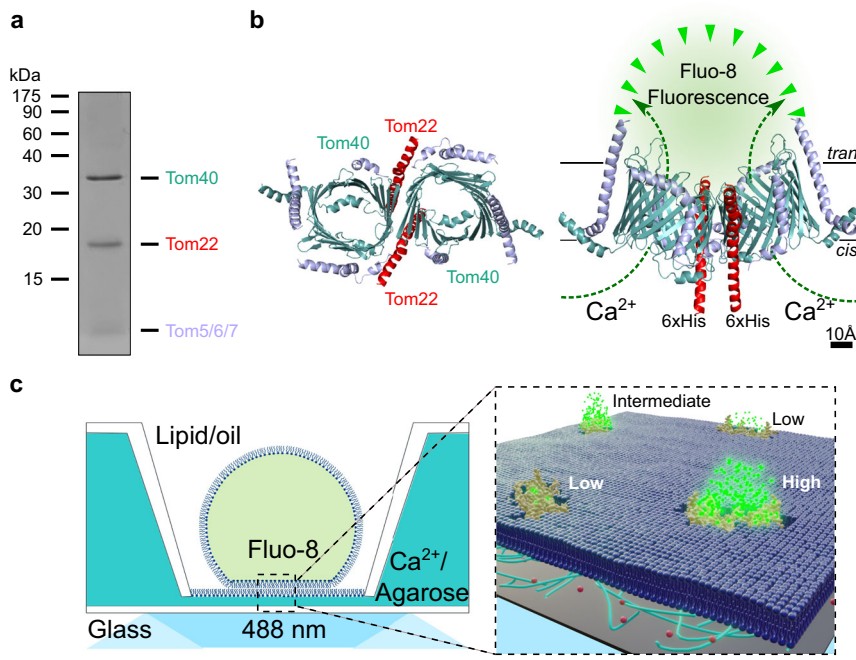

**Fig. 1 Scheme for tracking single TOM-CC molecules and imaging their ion channel activity. a** TOM-CC was isolated from mitochondria of a *Neurospora* strain carrying a Tom22 with a hexahistidinyl tag (6xHis). Analysis of purified protein by SDS-polyacrylamide gel electrophoresis (SDS-PAGE) followed by Coomassie Blue staining revealed all known subunits of the core complex, Tom40, Tom22, Tom7, Tom6 and Tom5. The small subunits Tom7, Tom6 and Tom5 are not separated by SDS-PAGE. **b** Atomic model based on the cryoEM map of *N. crassa* TOM core complex (EMDB, EMD-3761[6]). The ionic pathway through the two aqueous β-barrel Tom40 pores is used to optically study the open-closed channel activity of individual TOM-CCs. Left, cytosolic view; right, side view; *cis*, mitochondria intermembrane space; *trans*, cytosol. Tom7, Tom6 and Tom5 are not labeled for clarity. **c** Single-molecule tracking and channel activity sensing of TOM-CC in DIB membranes using electrode-free optical single-channel recording. Left: Membranes are created through contact of lipid monolayer-coated aqueous droplets in a lipid/oil phase and a lipid monolayer on top of an agarose hydrogel. The *cis* side of the membrane contained $Ca^{2+}$-ions (0.66 M), while having at the trans side a $Ca^{2+}$-sensitive fluorescent dye (Fluo-8) and KCl (1.32 M) to balance the membrane osmotic pressure. The high $Ca^{2+}$ content is necessary to induce high calcium flux through the TOM-CC pores and to provide reliable optical signals. Right: $Ca^{2+}$-ion flux through individual TOM-CCs from *cis* to *trans* is driven by a $Ca^{2+}$ concentration gradient, established around the two Tom40 pores, and measured by monitoring Fluo-8 emission in close proximity to the membrane using TIRF microscopy. Fluorescence signals reveal the local position of individual TOM-CCs, which is used to determine their mode of lateral diffusion in the membrane. The level of the fluorescence (high, intermediate and low intensity) correlates with corresponding permeability states of a TOM-CC molecule. A 100× TIRF objective is used both for illumination and imaging. Green dots, fluorescent Fluo-8; red dots, $Ca^{2+}$ ions.

**The permeability states of TOM-CC are coupled to lateral mobility.** Lateral mobility, e.g., free, restricted or directed diffusion, is an important factor for the organization and function of biological membranes and its components[23–28]. In some cases, the transient anchoring of mobile membrane proteins thereby precedes protein-induced signaling events, as shown for CFTR[29] and presynaptic calcium channels[30]. It is also well accepted that there is a direct correlation between activity and rotational membrane diffusion for cytochrome oxidase[31] and sarcoplasmic reticulum ATPase[32,33]. Since rotational diffusion is obligatorily coupled to lateral diffusion (both require a fluid crystalline membrane), we asked whether, and if so, how the mode of diffusion affects the channel activity of TOM-CC and its permeability states $S_H$, $S_I$, and $S_L$. To this end, we simultaneously tracked the open-closed activity and position of individual TOM-CC molecules in the membrane over time (Fig. 3a).

Supplementary Movie S3 and Fig. 3b show that the open-closed channel activity of single TOM-CCs is coupled to lateral movement in the membrane. This is supported by comparison of the trajectories of single TOM-CC molecules with their corresponding fluorescence amplitude traces (Fig. 3c, d). The position of fluorescent spots does not change when TOM-CC is in intermediate $S_I$ or low $S_L$ permeability state. Although weak intensity profiles do not allow accurate determination of the position of TOM-CC in the membrane plane, Supplementary

Movie S3 and Fig. 3d clearly show that TOM-CC does not move in $S_L$; disappearance and reappearance of the fluorescent spot, switching from $S_I$ to $S_L$ and back to $S_I$, occurs at virtually the same spatial x, y coordinates. In contrast, the trajectories of TOM-CC in $S_H$ state demonstrate free diffusion. Additional samples of trajectories and amplitude traces are shown in Supplementary Fig. S3.

Similar *stop-and-go* movement patterns were observed in an independent set of experiments for TOM-CC covalently labeled with fluorescent dye Cy3 (Supplementary Movie S4 and Fig. S8). It is particularly striking that freely moving TOM-CC molecules stop at the same spatial x, y-position when they cross the same position a second time, indicating a specific molecular trap or anchor point (agarose polysaccharide strands) at this stop-position below the membrane. In contrast, single Tom40 molecules isolated from TOM-CC (Supplementary Movie S5, Fig. 3e and Supplementary Fig. S4) and OmpF molecules purified from *E. coli*. outer membranes (Supplementary Movie S6 and Supplementary Fig. S5) show the most elementary mode of mobility expected for homogeneous membranes: simple Brownian translational diffusion. Consistent with the fact that OmpF is fully embedded in membranes[22], only very few out of a total of 171 analyzed OmpF molecules were temporarily (~15%) or permanently (~12%) trapped in the membrane (Supplementary Figs. S7, S6). In contrast to TOM-CC, however, a "stop" of OmpF

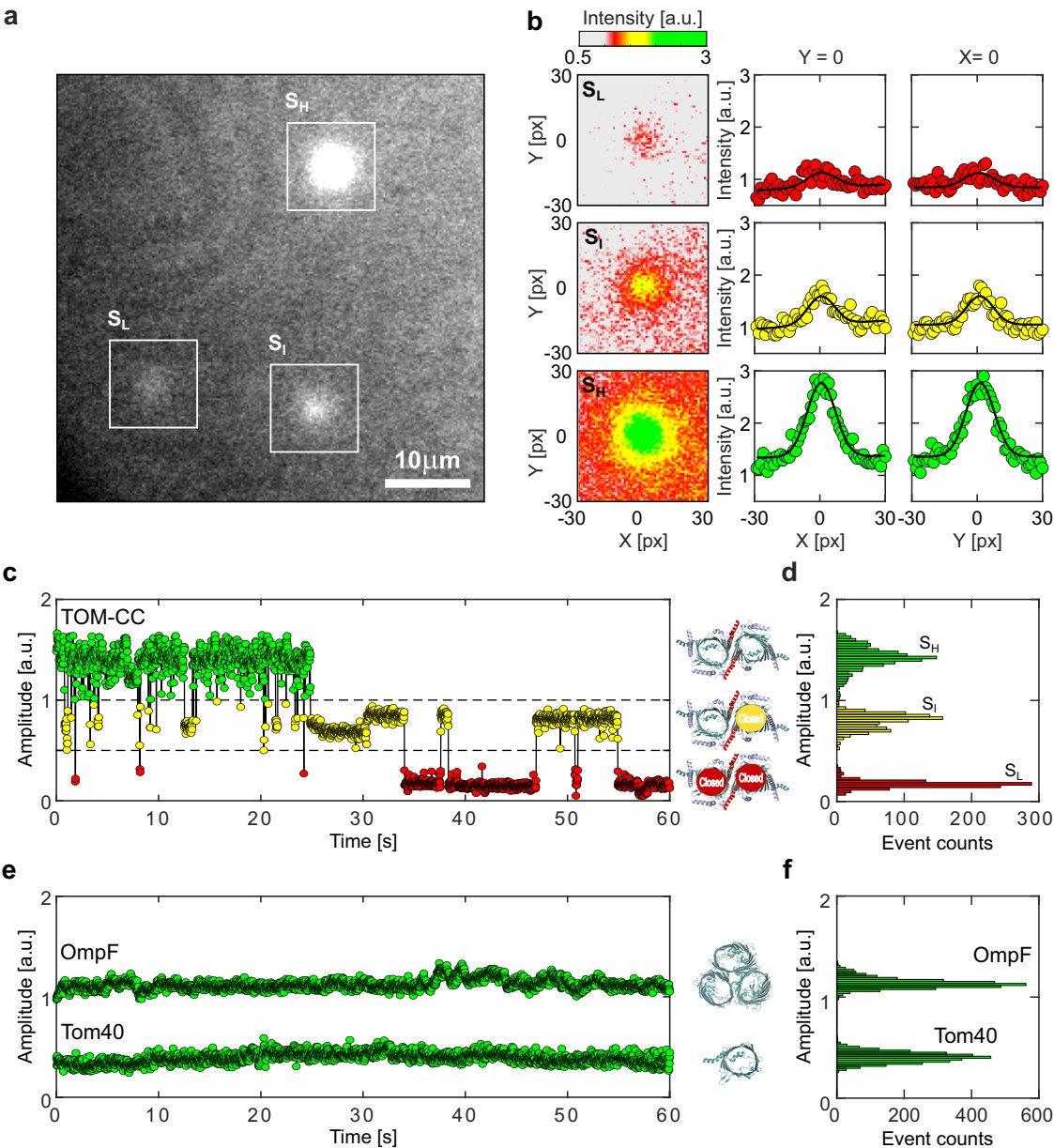

**Fig. 2 Visualizing the two-pore channel activity of TOM-CC. a** Typical image ($N > 5.3 \times 10^5$) of a non-modified agarose-supported DIB membrane with TOM-CC channels under 488 nm TIRF-illumination. The white squares mark spots of high ($S_H$), intermediate ($S_I$), and low ($S_L$) intensity (Supplementary Movie S1). The TIRF image has not been corrected by fluorescence bleaching or by a filter algorithm. **b** Fitting the fluorescence intensity profile of the three spots marked in (**a**) to two-dimensional Gaussian functions (Supplementary Movie S2). Red, yellow and green intensity profiles represent TOM-CC in $S_L$, $S_I$, and $S_H$ demonstrating Tom40 channels, which are fully closed, one and two channels open, respectively. Pixel size, 0.16 μm. **c** Fluorescence amplitude trace and (**d**) amplitude histogram of the two-pore β-barrel protein channel TOM-CC. The TOM-CC channel switches between $S_H$, $S_I$, and $S_L$ permeability states over time. Inserts, schematic of *N. crassa* TOM core complex with two pores open in $S_H$ (top), one pore open in $S_I$ (middle), and two pores closed in $S_L$ (bottom) (EMDB, EMD-3761[6]); **e** Representative single-channel optical recordings and (**f**) amplitude histograms of the TOM-CC subunit Tom40 and OmpF. Tom40 and OmpF are completely embedded in the lipid bilayer. In contrast to the two-pore β-barrel protein complex TOM-CC, Tom40, and OmpF exhibit one permeability state over time only. Insert, structural model of *N. crassa* Tom40 (EMDB, EMD-3761) and *E. coli* OmpF (PDB, 1OPF) with open β-barrel pores. Data were acquired as described in Fig. 1c at a frame rate of 47.5 s$^{-1}$. a.u. arbitrary unit.

did not seem to be accompanied by a change in intensity (Supplementary Figs. S6, S7) and thus by closing the pores.

In good agreement with these results, the diffusion coefficients of the TOM-CC, evaluated from the activity profiles and trajectories (Fig. 3 and Supplementary Fig. S3), were determined from time-averaged mean squared displacements as $D(S_I) = D(S_L) \leq D_{min} = 0.01 \mu m^2 s^{-1}$ and $D(S_H) \simeq 0.85 \pm 0.16$ (mean ± *SEM*, $n = 46$) in states $S_I$ and $S_H$, respectively. TOM-CC molecules, that revealed diffusion constants less or equal than $D_{min}$, were defined as

immobilized. The diffusion coefficient $D(S_H)$ corresponds to the typical values of mobile Tom40 ($D_{Tom40} \sim 0.5 \mu m^2 s^{-1}$) and mobile Tom7 ($D_{Tom7} \sim 0.7 \mu m^2 s^{-1}$) in native mitochondrial membranes[34,35] and is comparable to that of transmembrane proteins in plasma membranes lined by cytoskeletal networks[28]. The diffusion coefficient $D(S_L)$ of TOM-CC in $S_L$ state could not always be reliably determined due to its extremely low intensity levels.

TOM-CC labeled with Cy3 yielded diffusion constants of $D_{Cy3} \simeq 0.36 \pm 0.08 \mu m^2 s^{-1}$ (mean ± SEM, $n = 15$) and $D_{Cy3} \leq$

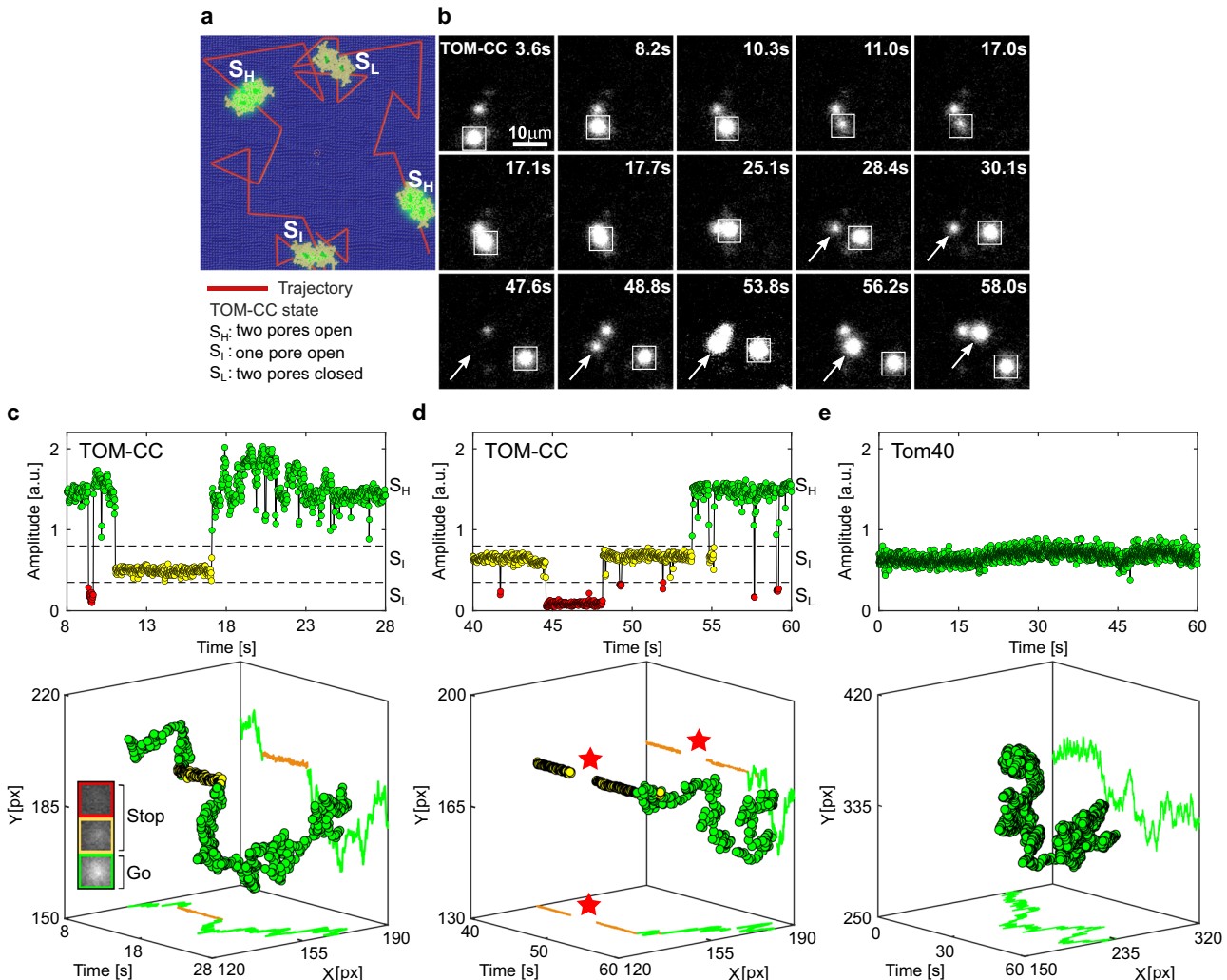

**Fig. 3 Lateral mobility correlates with the channel activity of TOM-CC. a** Scheme for imaging both position and channel activity of single TOM-CCs.
**b** Representative TIRF microscopy images of a non-modified agarose-supported DIB membrane with three TOM-CC molecules taken from a time series of
60 s. The square-marked spot displays lateral motion, interrupted by a transient arrest between $t = 11.0$ s and $t = 17.0$ s. The arrow-marked spot
corresponds to a non-moving TOM-CC until $t = 48.8$ s. Afterwards, it starts moving. Both moving spots show high fluorescence intensity ($S_H$); the non-
moving spots display intermediate ($S_I$) or dark ($S_L$) fluorescence intensity (Supplementary Movie S3). **c**, **d** Fluorescent amplitude trace and corresponding
trajectory of the square- and arrow-marked TOM-CC as shown in (**b**) highlighted for two different time windows. Plots on top shows the change of
amplitude over time, and plots on the bottom show the respective spatiotemporal dynamics for the three states. Comparison of the trajectories of single
TOM-CC molecules with their corresponding amplitude traces reveals a direct correlation between *stop-and-go* movement and open-closed channel
activity. Lateral diffusion of TOM-CCs in the DIB membrane is interrupted by temporary arrest, presumably due to transient linkage to the underlying
agarose hydrogel. Although weak intensity profiles in $S_I$ do not allow accurate position determination, the fluorescent spots disappear and reappear at the
same spatial x, y coordinates (red stars). The higher amplitude (**c** top) between $t = 17.1$ s and $t = 25.1$ s is due to the overlap between two adjacent spots.
Green, TOM-CC is freely diffusive in $S_H$; yellow and red, immobile TOM-CC in $S_I$ and $S_L$. **e** Fluorescent amplitude trace and corresponding trajectory of a
single β-barrel subunit Tom40. The Tom40 channel exhibits only one permeability state and is subject to simple thermal movement in the membrane
(Supplementary Movie S5). Tom40 does not show *stop-and-go* motion as with TOM-CC. All data were acquired as described in Fig. 1c at a frame rate of
47.5 s$^{-1}$. A total of $n_{TOM} = 64$ and $n_{Tom40} = 20$ amplitude traces and trajectories were analyzed. a.u., arbitrary unit.

$D_{min} = 0.01 \, \mu m^2 \, s^{-1}$ for moving and transiently trapped particles,
respectively (Supplementary Fig. S8). In agreement with typical
values for monomeric and multimeric proteins in homogenous
lipid membranes[36,37], the lateral diffusion coefficients of the
control proteins Tom40 and OmpF yielded translational diffusion
coefficients of $D_{Tom40} \simeq 1.49 \pm 0.21 \, \mu m^2 \, s^{-1}$ (mean ± SEM, $n = 20$)
and $D_{OmpF} \simeq 1.16 \pm 0.07 \, \mu m^2 \, s^{-1}$ (mean ± SEM, $n = 42$), respec-
tively (Supplementary Figs. S4, S5).

Based on these results, we concluded that the arrest of TOM-
CC in the lipid bilayer membrane, caused by short-lived
interaction with the polysaccharide strands of the supporting

hydrogel (Supplementary Fig. S1), triggers a transient closure of
its two β-barrel pores.

**Controlled immobilization of the TOM-CC results in channel
closures.** Since the two Tom22 subunits in the middle of the TOM-
CC (Fig. 1b) clearly protrude from the membrane plane at their
intramembrane space (IMS) side[6–9], we considered whether Tom22
acts as "light-switch" that determines the lateral mobility of the
TOM-CC and thereby causing transitions between open ($S_H$) and
the closed ($S_I$ and $S_L$) conformations of the two TOM-CC pores.

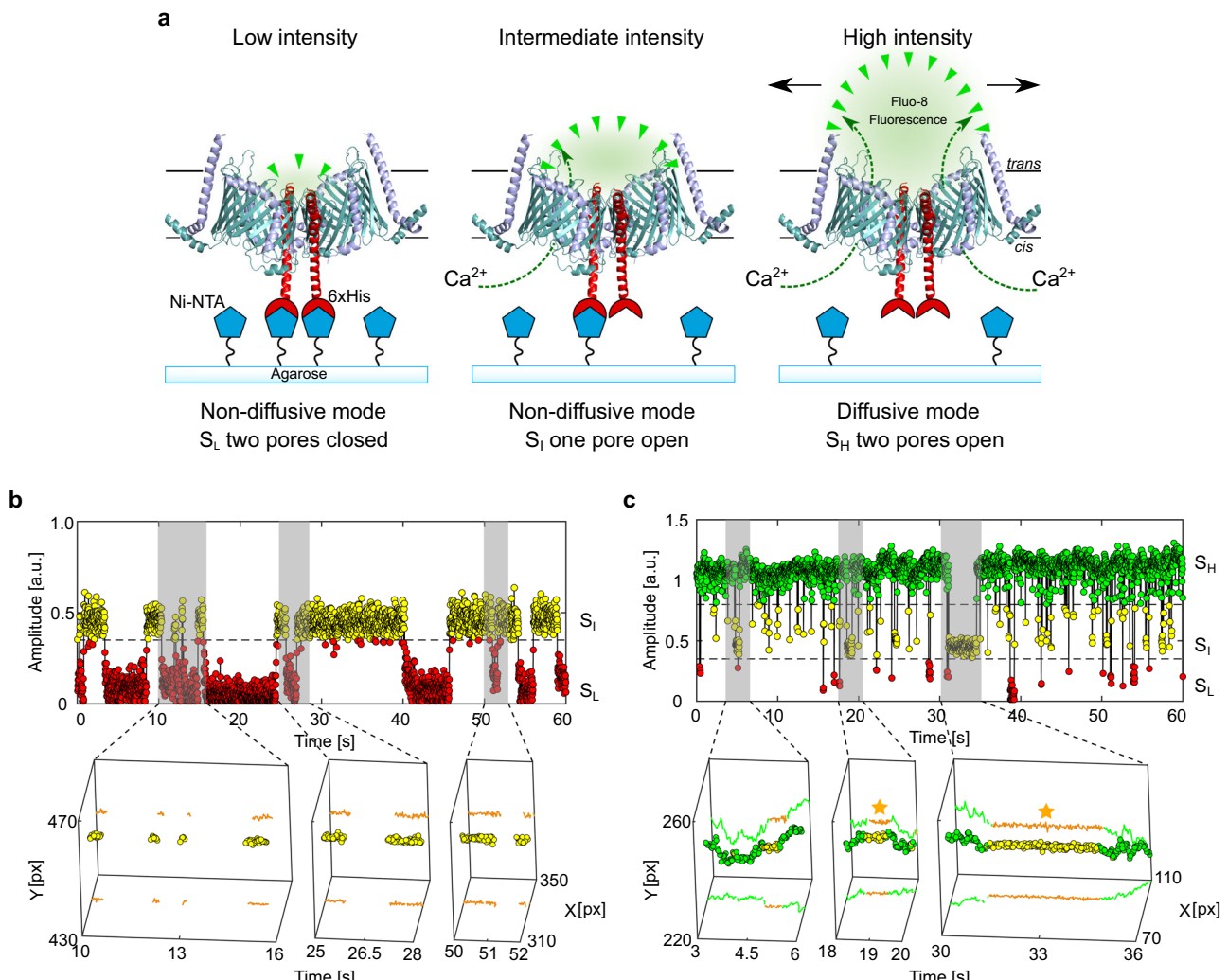

**Fig. 4 Controlled immobilization of TOM-CC triggers channel closures. a** Schematic representation of individual TOM-CC channels in DIB membranes supported by Ni-NTA-modified agarose. TOM-CC molecules can be permanently linked to the underlying hydrogel via His-tagged Tom22. Tethered and non-tethered TOM-CC molecules in closed ($S_I$ and $S_L$) and open ($S_H$) states are indicated, respectively. **b** Fluorescent amplitude trace (top) of a TOM-CC channel permanently tethered to Ni-NTA-modified agarose. The trajectory segments (bottom) correspond to the time periods of the amplitude traces marked in gray (Supplementary Movie S7: bottom). Non-diffusive, permanently immobilized TOM-CC is only found in $S_I$ or $S_L$, indicating that tight binding of the His-tagged Tom22 domain (Fig. 1b) to Ni-NTA-modified agarose triggers closure of the β-barrel TOM-CC pores. **c** Fluorescence amplitude trace (top) of a TOM-CC channel transiently and non-specifically entangled by Ni-NTA-modified agarose. The trajectory segments (bottom) correspond to the time periods of the amplitude traces marked in gray. The movement of TOM-CC is interrupted twice at the same spatial $x,y$ membrane position from $t_1 = 18.56$ s to $t_2 = 19.19$ s and from $t_3 = 31.14$ s to $t_4 = 34.55$ s (yellow stars). Consistent with the data shown in Fig. 3, moving TOM-CC molecules in diffusive mode are found in the fully open $S_H$ state; transient tethering causes the TOM-CC β-barrels to close. Data were acquired as described in Fig. 1c at a frame rate of $47.5$ s$^{-1}$. A total of $n_{TOM} = 123$ amplitude traces and trajectories were analyzed. a.u. arbitrary unit.

To address this question, we employed a Ni-NTA-modified agarose to permanently restrict lateral movement of the protein by fixing single TOM-CC molecules to the hydrogel via the C-terminus of His-tagged Tom22 (Fig. 4a, Supplementary Movie S7).

As expected, permanently immobilized TOM-CC ($D(S_I)$ and $D(S_L) \leq D_{min} = 0.01 \ \mu m^2 \ s^{-1}$, $n = 83$) was most often found in states $S_I$ or $S_L$, indicating one or two pores closed, respectively (Fig. 4b, Supplementary Movie S7 and Fig. S9a–d). Only a minority of TOM-CC molecules that were not immobilized by the binding of Tom22 to Ni-NTA moved randomly in the membrane plane ($D(S_H) \simeq 0.34 \pm 0.06 \ \mu m^2 \ s^{-1}$, mean $\pm$ SEM, $n = 40$). The movement of this latter population of molecules was occasionally interrupted by periods of transient anchorage ($D(S_I)$ and $D(S_L) \leq D_{min}$), as observed for molecules in

membranes supported by unmodified agarose. Again, moving TOM-CC molecules were found in the fully open $S_H$ state; non-moving complexes in the $S_I$ or the $S_L$ states (Fig. 4c; Supplementary Fig. S9e–h). For comparison, purified Tom40 itself showed only one ion permeation state and no *stop-and-go* dynamics (Fig. 3e, Supplementary Movie S5 and Supplementary Fig. S4). Consistently, in the presence of imidazole, which prevents binding of hexahistidine-tagged Tom22 to Ni-NTA-modified agarose (Supplementary Movie S8), also virtually no permanently immobilized TOM-CC molecules were observed ($D(S_H) \simeq 1.35 \pm 0.14 \ \mu m^2 \ s^{-1}$, mean $\pm$ SEM, $n = 30$).

**Correlation between lateral motion and TOM-CC channel activity.** DIB membranes supported by both hydrogels, non-modified and Ni-NTA-modified agarose (Supplementary Movie S7), showed a

significant number of single TOM-CC molecules that were either non-diffusive or diffusive at $S_I$ and $S_H$. Thus, TOM-CC molecules were numerically sorted into diffusive $\{D > D_{min}\}$ and non-diffusive $\{D \leq D_{min}\}$ tethered groups to emphasis the correlation between the mode of lateral diffusion and channel activity of 187 observed TOM-CC molecules. We then asked whether the molecules were in the fully opened $S_H$ or half-open $S_I$ state. Since weak intensity profiles did not allow accurate determination of the position of TOM-CC and thus the lateral diffusion coefficient $D(S_L)$, we conducted the classification analysis using only the parameters $D(S_H)$ and $D(S_I)$.

As shown in Fig. 5a, b, we can define three different classes of lateral motion and channel activity. The first and major class I $\{I | D(S_H) > D_{min} \cap D(S_I) \leq D_{min}\}$ shows lateral mobility at $S_H$ only, while being tethered at $S_I$. Here, spatiotemporal *stop-and-go* dynamics of the TOM-CC correlate with channel activity. The second class II $\{II | D(S_H) > D_{min} \cap D(S_I) > D_{min}\}$ shows similar diffusivities at both states, $S_H$ and $S_I$ (Supplementary Fig. S10). The TOM molecules in this class are unlikely to have a functional Tom22[21,38] or are incorporated into the membrane in a reverse orientation and therefore do not interact with the hydrogel. Another possible but unlikely explanation is a spatial void of agarose network preventing mechanoregulated interaction of Tom22 with the network within the observation time window. The third class III $\{III | D(S_H) \leq D_{min} \cap D(S_I) \leq D_{min}\}$ represents events exhibiting permanently tethered TOM-CC molecules, which do not move in the membrane and are exclusively non-diffusive (Supplementary Fig. S9a, b). Most molecules in this class are in $S_I$ and $S_L$. Those TOM-CC molecules, which briefly change from $S_I$ to $S_H$ and back to $S_I$ (Supplementary Fig. S9c, d), could diffuse in $S_H$ but are immediately recaptured and trapped by the hydrogel below the membrane, resulting in $D(S_H) \leq D_{min}$.

Figure 5c shows state probabilities of TOM-CC in membranes supported by the two different hydrogels, non-modified and Ni-NTA-modified agarose. Diffusive molecules $(D > D_{min})$ in membranes supported both by non-modified and Ni-NTA-modified agarose show similar probabilities to be at one of the three permeability states ($S_H$, $S_I$, and $S_L$). Diffusive TOM-CC molecules are significantly more often at $S_H$ than at $S_I$ and $S_L$. The permanently tethered fraction of TOM-CC (67%) in Ni-NTA-modified agarose is ~2.4 times larger compared to the fraction (28%) in non-modified agarose, consistent with the stronger interaction of Tom22 with the hydrogel, thereby permanently constraining lateral motion. In line with this, permanently tethered molecules $(D \leq D_{min})$ in both hydrogel-supported membranes stay at $S_I$ during the majority of time, and show only transient $S_H$ and $S_L$ occupancy. The data suggest that the C-terminal (IMS) domain of Tom22 plays a previously unrecognized role in mechanoregulation of TOM-CC channel activity by binding to immobile structures near the membrane.

Although diffusive TOM-CC molecules $(D > D_{min})$ are observed more often at $S_H$ in Ni-NTA-modified agarose than in non-modified agarose-supported membranes (Fig. 5c), they show a lower stability at $S_H$ having a significantly higher transition probability for switching between $S_H$ and $S_I$ (Fig. 5d, ($S_H \leftrightarrows S_I$) $\simeq$ 5.3% versus ($S_H \leftrightarrows S_I$) $\simeq$ 2.3%). This is in line with the higher efficacy of TOM-CC-trapping by Ni-NTA-modified agarose compared to non-modified agarose. In contrast to unmodified hydrogel, Ni-NTA-modified agarose hydrogel can capture freely mobile TOM-CC via the (IMS) domain of Tom22 in two ways: on the one hand, by specific interaction and permanent anchoring with Ni-NTA, and, on the other hand, by collision and transient nonspecific anchoring. While a direct transition between $S_H$ and $S_L$ barely occurs in both systems, transitions between $S_I$ and $S_L$ are similarly often. This indicates that the two Tom40 β-barrel pores independently open and close within the time resolution (~20 ms) of our experiment.

## Discussion

In this study, we have shown that TOM-CC molecules can interact with their environment and switch reversibly between three states of transmembrane channel activity. Only when both channels are open (high permeable state) the TOM-CC molecules move freely in the membrane, while transient physical interaction with the hydrated agarose film supporting the membrane leads to temporary immobilization and closure of at least one pore. The use of agarose films with covalently bound Ni-NTA promotes TOM-CC immobilization. Duration of channel closure is thereby significantly prolonged. This indicates that TOM-CC does not only respond to biochemical[39], but also to mechanical stimuli: the anchoring of freely moving TOM-CC to structures near the membrane leads to a partial or complete closure of the two-pore TOM-CC channel.

We obtained data from $n = 187$ single molecules and over half a million image frames (Fig. 5b, $N_{TOM-CC} = 532,576$). This sampling number allows us to assign molecular events with statistical confidence using non-parametric statistics, but even visual perusal of the data is sufficient to allow a first assessment of data reliability. Our data is consistent with the dimeric pore structure of the TOM-CC where the three molecular states correspond to one or two pores open and all pores closed, respectively. At least in our model system recent proposals of a trimeric functional pore structure[40] are not consistent with our data. Functional reversible disassembly and reassembly of the TOM-CC at time scales studied here are highly improbable.

Previous electrophysiology studies[41] have shown that the TOM channel of yeast mitochondria is mainly in a closed state, while TOM lacking Tom22 has been found mainly in an open state, similar to that of yeast and *N. crassa* Tom40[11,21,41]. It has therefore been postulated that Tom22 negatively regulates the opening probability of TOM[41]. However, the molecular mechanism of how Tom22 influences the activity of the open-closed channels of the TOM machinery has remained an open question.

Examination of the cryoEM structures of TOM-CC reveals that its two Tom22 subunits extend significantly into the IMS space (more than 22 Å[6–9], Fig. 1b), thereby preventing a direct interaction of the two Tom40 β-barrels with the agarose matrix. It is therefore likely that the two Tom22 subunits localized in the center of the complex are primarily responsible for the matrix-dependent channel activity. This is supported by our observation that binding of one hexahistidine-tagged Tom22 to the Ni-NTA matrix leads to a permanent immobilization of the TOM-CC in the membrane, with one of the two Tom40 pores open. Restricting the movement of the second inter-membrane space domain of Tom22 could then lead to the closure of the second Tom40 pore. Consistent with this, the isolated Tom40 pore itself showed essentially no correlated *stop-and-go* and open-closed dynamics.

In conclusion, we have clear evidence—structural and dynamic—that the two Tom22 subunits of TOM-CC can force the Tom40 dimer to undergo a conformational change that leads to channel closure. This process is reversible and triggered solely by natural thermal fluctuations of the TOM-CC in the membrane. We realize that the agarose matrix underlying the membrane is not a perfect substitute for the intermembrane space of mitochondria. Nevertheless, the reconstituted system allows the complex effects of mitochondrial compartmentalization to be eliminated as shown for plasma membrane proteins localized in polymer supported lipid bilayers[20,42].

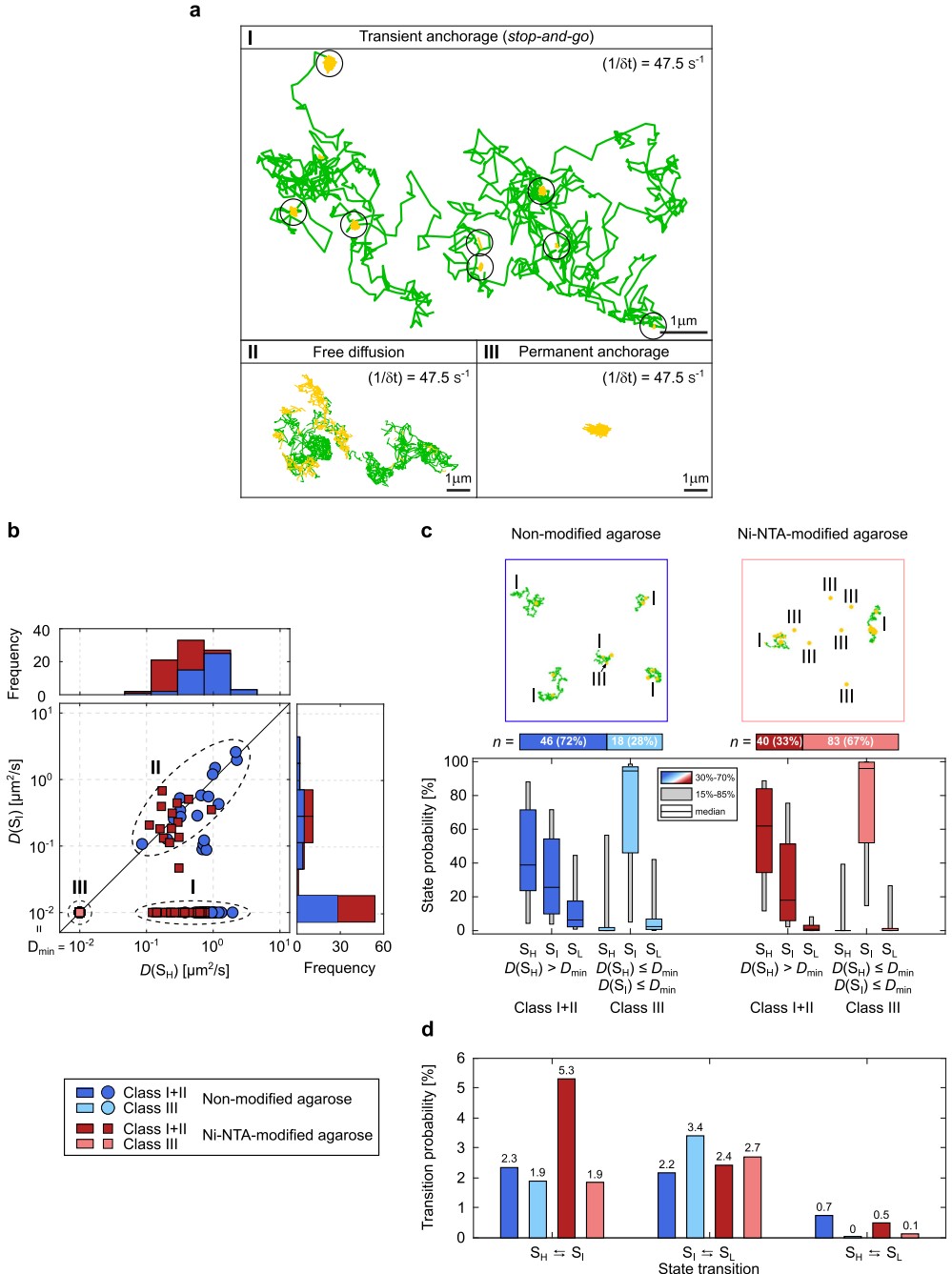

**Fig. 5 Statistical correlation between channel activity and lateral mobility of TOM-CC. a** Trajectories for various modes of TOM-CC mobility. Class I, diffusion interrupted by periods of transient anchorage (sites of transient anchorage circled); Class II, free diffusion; Class III, permanent anchorage. Moving particles in $S_H$ are shown in the trajectories in green; transiently or permanently tethered molecules in $S_I$ are shown in yellow. Data were acquired for $t = 60$ s at a frame rate of 47.5 s$^{-1}$. **b** $D(S_I)$ as a function of $D(S_H)$ individually plotted for all TOM-CC molecules in DIB membranes supported by non-modified (dark blue and light blue, $n = 64$) and Ni-NTA modified agarose (dark red and light red, $n = 123$). Frequency histograms of $D(S_I)$, $D(S_H)$ and $D(S_I)$ are shown on top and right side, respectively. Three classes can be defined: a main class (I) of moving particles in $S_H$ while being transiently tethered at $S_I$ $\{I | D(S_H) > D_{min} \cap D(S_I) \leq D_{min}\}$, a second class (II) of freely moving particles in $S_H$ and $S_I$ $\{II | D(S_H) > D_{min} \cap D(S_I) > D_{min}\}$ and a third class (III) of permanently tethered molecules in $S_I$ and $S_L$ $\{III | D(S_H) \leq D_{min} \cap D(S_I) \leq D_{min}\}$. **c** Example trajectories (top and Supplementary Movie S7) and state probabilities (bottom) of non-permanently and permanently tethered TOM-CC in DIBs supported by non-modified and Ni-NTA-modified agarose. The probability of being in state $S_H$ is higher for non-permanently (classes I and II, dark blue [$n = 46$] and dark red [$n = 40$]) than for permanently tethered molecules (class III, light blue [$n = 18$] and light red [$n = 83$]). The probability of being in state $S_I$ is significantly higher for permanently (class III) than for non-permanently tethered particles (classes I and II). This suggests that binding of Tom22 to Ni-NTA agarose below the membrane triggers closure of the TOM-CC channels. The data are represented as median; the confidence intervals are given between 15 to 85% and 30 to 70%. **d** Absolute state transition probabilities classified by bidirectional state transitions as $S_H \leftrightarrows S_I$, $S_I \leftrightarrows S_L$, and $S_H \leftrightarrows S_L$. Diffusive TOM-CC molecules have a significantly higher transition probability for switching between $S_H$ and $S_I$ in DIBs supported by Ni-NTA-modified agarose (~5.3%) than in non-modified agarose membranes (~2.3%). This is consistent with the higher efficacy of TOM-CC-trapping by Ni-NTA-modified agarose compared to non-modified agarose. Classification of non-permanently and permanently tethered TOM-CC is shown at the left bottom.

Our results raise a fundamental question: are the two Tom40 channels open continually, as static cryoEM structures seem to indicate[6–9], or can the protein channels be actively influenced by interaction with exogenous interacting proteins near the membrane that restrict the movement of the TOM-CC? In the future, the development of improved in vitro assays may help to observe intrinsic TOM-CC dynamics and could reveal even more processes and dynamics of possible physiological relevance. With regard to the mechanism of TOM-mediated protein translocation into mitochondria, our data open the exciting possibility that proteins at the periphery of the outer membrane, e.g., components of inner-membrane TIM23 complex, not only limit lateral membrane diffusion of the TOM-CC by transient interaction and formation of an active supercomplex at mitochondrial contact sites[43], but might also regulate the open-close status of the channel as a consequence. This could in turn translate into conformational changes of the TOM-CC that influence the translocation process of incoming mitochondrial preproteins. In this context, it might be interesting to visualize different Tom22-dependent conformations of TOM-CC using cryoEM and to perform single-molecule FRET experiments to study the dynamics of TOM-CC between different conformations in the near future.

Recently, the now extensive structural and physical data common to mechanosensitive membrane proteins have been reviewed[44,45]. The structural data now allows mechanosensitive proteins and protein complexes (all of which were comprised of transmembrane α-helices) to be separated into five different classes, each subject to characteristic underlying molecular mechanisms. They also show that for the membrane channels considered, fundamental physical properties of the membrane can influence channel activity. This ancient mechanism to regulate open-closed channel activity has also been observed in MSL1, a mitochondrial ion channel that dissipates the mitochondrial membrane potential to maintain redox homeostasis during abiotic stress[46–49].

In this study, we provide in vitro evidence for the first example of a membrane β-barrel protein complex that exhibits membrane state-dependent mechanosensitive-like properties. Our findings are consistent with the "tether model"[45], where an inter-membrane protein anchor domain (here Tom22) limits lateral diffusion, as observed for a large number of α-helical membrane proteins[44,45,50–53]. Since the TOM-CC complex bends the outer mitochondrial membrane locally towards the intermembrane space with an average radius of about 140 Å[54], the anchor domains of the two Tom22 subunits should easily be able to dock with other proteins in the mitochondrial intermembrane space and the mitochondrial inner membrane. It will be interesting to study as to whether the mechanostimulated conformational change of the TOM-CC observed in vitro can be confirmed in intact mitochondria.

## Methods

### Growth of *Neurospora crassa* and preparation of mitochondria. *Neurospora crassa* (strain GR-107) that contains a hexahistidinyl-tagged form of Tom22 was grown and mitochondria were isolated as described[6]. Briefly, ~1.5 kg (wet weight) of hyphae were homogenized in 250 mM sucrose, 2 mM EDTA, 20 mM Tris pH 8.5, 1 mM phenylmethylsulfonyl fluoride (PMSF) in a Waring blender at 4 °C. ~1.5 kg of quartz sand was added and the cell walls were disrupted by passing the suspension through a corundum stone mill. Cellular residues were pelleted and discarded in two centrifugation steps ($4000 \times g$) for 5 min at 4 °C. The mitochondria were sedimented in 250 mM sucrose, 2 mM EDTA, 20 mM Tris pH 8.5, 1 mM PMSF at $17,000 \times g$ for 80 min. This step was repeated to improve the purity. The isolated mitochondria were suspended in 250 mM sucrose, 20 mM Tris pH 8.5, 1 mM PMSF at a final protein concentration of 50 mg/ml, shock-frozen in liquid nitrogen and stored at −20 °C.

### Isolation of TOM core complex. TOM-CC, containing subunits Tom40, Tom22, Tom7, Tom6, and Tom5, were purified from *Neurospora crassa* strain GR-107 as

described[6,10,19]. *N. crassa* mitochondria were solubilized at a protein concentration of 10 mg/ml in 1% (w/v) n-dodecyl-β-D-maltoside (Glycon Biochemicals, Germany), 20% (v/v) glycerol, 300 mM NaCl, 20 mM imidazole, 20 mM Tris-HCl (pH 8.5), and 1 mM PMSF. After centrifugation at $130,000 \times g$, the clarified extract was loaded onto a nickel-nitrilotriacetic acid column (Cytiva, Germany). The column was rinsed with the same buffer containing 0.1% (w/v) n-dodecyl-β-D-maltoside and TOM core complex was eluted with buffer containing 0.1% (w/v) n-dodecyl-β-D-maltoside, 10% (v/v) glycerol, 20 mM Tris (pH 8.5), 1 mM PMSF, and 300 mM imidazole. For further purification, TOM core complex containing fractions were pooled and loaded onto a Resource Q anion exchange column (Cytiva) equilibrated with 20 mM Hepes (pH 7.2), 2% (v/v) dimethyl sulfoxide (DMSO) and 0.1% (w/v) n-dodecyl-β-D-maltoside. TOM core complex was eluted with 0–500 mM KCl. A few preparations contained additional phosphate (~0.19 mM). The purity of protein samples (0.4–1.2 mg/ml) was assessed by sodium dodecyl sulfate polyacrylamide gel electrophoresis (SDS-PAGE) followed by staining with Coomassie Brilliant Blue.

### Fluorescence labeling of TOM core complex. TOM-CC was covalently labeled with the fluorescent dye Cy3 according to Joo and Ha[55]. Briefly, about 1 mg/ml of purified TOM-CC was reacted with Cy3-maleimide (AAT Bioquest, USA) at a molar ratio complex to dye of 1:5 in 20 mM HEPES (pH 7.2), 2% (v/v) dimethyl sulfoxide, 350 mM KCl and 0.1% (w/v) n-dodecyl-β-D-maltoside at 25 °C for 2 h in the dark. Labeled protein was separated from unconjugated dye by affinity chromatography using Ni-NTA resin, subjected to SDS-PAGE and visualized by 555 nm light and Coomassie Brilliant Blue staining.

### Isolation of Tom40. For the isolation of Tom40, isolated mitochondria of *N. crassa* strain GR-107 were solubilized at a protein concentration of 10 mg/ml in 1% (w/v) *n*-dodecyl β-D-maltoside (Glycon Biochemicals, Germany), 20% (v/v) glycerol, 300 mM NaCl, 20 mM imidazole, 20 mM Tris (pH 8.5), and 1 mM PMSF for 30 min at 4 °C[21]. After centrifugation at $130,000 \times g$, the clarified extract was filtered and loaded onto a Ni-NTA column (Cytiva, Germany). The column was rinsed with 0.1% DDM, 10% glycerol, 300 mM NaCl, and 20 mM Tris (pH 8.5) and Tom40 was directly eluted with 3% (w/v) *n*-octyl β-D-glucopyranoside (OG; Glycon Biochemicals, Germany), 2% (v/v) DMSO, and 20 mM Tris (pH 8.5). The purity of the isolated protein (~0.3 mg/ml) was assessed by SDS-PAGE.

### Isolation of OmpF. Native OmpF protein was purified from *Escherichia coli* strain BE BL21(DE3)omp6, lacking both LamB and OmpC as described[56]. Cells from 1 L culture were suspended in 50 mM Tris-HCl, pH 7.5 buffer containing 2 mM MgCl₂ and DNAse and broken by passing through a French press. Unbroken cells were removed by a low-speed centrifugation, then, the supernatant was centrifuged at $100,000 \times g$ for 1 h. The pellet was resuspended in 50 mM Tris-HCl, pH 7.5, and mixed with an equal volume of SDS buffer containing 4% (w/v) sodium dodecyl sulfate (SDS), 2 mM β-mercaptoethanol and 50 mM Tris-HCl, pH 7.5. After 30 min incubation at a temperature of 50 °C, the solution was centrifuged at $100,000 \times g$ for 1 h. The pellet was suspended in 2% (w/v) SDS, 0.5 M NaCl, 50 mM Tris-HCl, pH 7.5, incubated at a temperature of 37 °C for 30 min and centrifuged again at $100,000 \times g$ for 30 min. The supernatant containing OmpF was dialyzed overnight against 20 mM Tris, pH 8, 1 mM EDTA and 1% (w/v) n-octyl polyoxyethylene (Octyl-POE, Bachem, Switzerland). The purity of the protein was assessed by SDS-PAGE.

### Formation of droplet interface bilayers. Droplet interface bilayer (DIB) membranes were prepared as previously described[17,18] with minor modifications. Glass coverslips were washed in an ultrasonic bath with acetone. Then the coverslips were rinsed several times with deionized water and dried under a stream of nitrogen. Subsequently, the glass coverslips were subjected to plasma cleaning for 5 min. 140 µl of molten 0.75% (w/v) low melting non-modified agarose ($T_m < 65 °C$, Sigma-Aldrich) or alternatively low melting Ni-NTA-modified agarose (Cube Biotech, Germany) was spin-coated at 5,000 rpm for 30 s onto the plasma-cleaned side of a glass coverslip. After assembly of the coverslip in a custom-built DIB device, the hydrogel film was hydrated with 2.5% (w/v) low melting agarose, 0.66 M CaCl₂ and 8.8 mM HEPES (pH 7.2) or with 2.5% (w/v) low melting agarose, 0.66 M CaCl₂, 300 mM imidazole and 8.8 mM HEPES (pH 7.2) and covered with a lipid/oil solution containing 9.5 mg/ml 1,2-diphytanoyl-sn-glycero-3-phosphocholine (DPhPC, Avanti Polar Lipids, USA) and 1:1 (v/v) mixture oil of hexadecane (Sigma-Aldrich) and silicon oil (Sigma-Aldrich). Aqueous droplets (~200 nl) containing 7 µM Fluo-8 sodium salt with a maximum excitation wavelength of 495 nm (Santa Cruz Biotechnology, USA), 400 µM EDTA, 8.8 mM HEPES (pH 7.2), 1.32 M KCl and ~2.7 nM TOM core complex or ~2 nM OmpF were pipetted into the same lipid/oil solution in a separate tray using a Nanoliter 2000 injector (World Precision Instruments). After 2 h of equilibration at room temperature, the droplets were transferred into the custom-built DIB device to form stable lipid bilayers between the droplet and the agarose hydrogel.

### AFM imaging. AFM images of agarose-coated coverslips were acquired in liquid solution using a Nanoscope 5 Multimode-8 AFM system with SNL-10 cantilevers (Bruker) in tapping mode.

**TIRF microscopy and optical recording**. An inverted total internal reflection fluorescence microscope (Ti-E Nikon) was used to image DIB membranes under TIRF illumination using a 488 nm laser (Visitron). Fluorescent emission of Fluo-8, transmitted through a Quad-Band TIRF-Filter (446/523/600/677 HC, AHF), was collected through a 100 × oil objective lens (Apochromat N.A. 1.49, Nikon) and recorded by a back-illuminated electron-multiplying CCD camera (iXon Ultra 897, 512 × 512 pixels, Andor) for 1 min at a frame rate of 47.51 s$^{-1}$. The pixel size was 0.16 μm.

**Tracking of fluorescence spots**. For reliable tracking and analysis of the spatiotemporal dynamics of individual fluorescence TOM-CC, Tom40 and OmpF channel activities a customized fully automated analysis routine was implemented in Matlab (The Mathworks, USA). The effect of bleaching was corrected by applying a standard fluorescence bleaching correction procedure[57]. A double exponential decay obtained by least-square fitting of the image series (average frame intensity $\langle I(t)_{raw}\rangle$) was obtained as

$$f(t, a_k) = a_1 \exp(a_2 t) + a_3 \exp(a_4 t) \tag{1}$$

where $t$ is the frame index (time) and $a_k$ are the fitting parameters. The time series was corrected according to

$$I(t) = I_{raw}(t)/f(t, a_k) \tag{2}$$

where $I(t)$ is the intensity as a function of time $t$. No filter algorithm was applied. The initial spatial position of a fluorescence spot was manually selected and within a defined region of interested (ROI, 30 × 30 pixels) fitted to a two-dimensional symmetric Gaussian function with planar tilt that accounts for possible local illumination gradients, that the global bleach correction cannot account of, as follows

$$G_{2D}(\boldsymbol{x}, \boldsymbol{\mu}, p_k) = p_1 + p_{2,3}(\boldsymbol{x} - \boldsymbol{\mu}) + A \exp(-(\boldsymbol{x} - \boldsymbol{\mu})^2/2\sigma^2) \tag{3}$$

where $\boldsymbol{x} = (x, y)$ is the ROI with the fluorescence intensity information, A and σ are the amplitude and width of the Gaussian, $p_k$ are parameters that characterize the background intensity of the ROI, and $\boldsymbol{\mu} = (x_0, y_0)$ defines the position of the Gaussian. The latter was used to update the position of ROI for the next image. Spots that temporal fuse their fluorescence signal with closely located spots were not considered due to the risk of confusing those spots.

**Data analysis**. The extracted amplitudes were separated by two individually selected amplitude-thresholds that dived the three states of activity ($S_H$, $S_I$, and $S_L$). The lateral diffusion constants $D(S_H)$ and $D(S_I)$ were obtained individually for spots within the respective high and intermediate amplitude range by linear regression of the time delay $\tau$ and the mean square displacement of the spots as

$$D = \frac{\langle|\boldsymbol{r}(x, y, \tau) - \boldsymbol{r}(x_0, y_0, \tau)|^2\rangle}{4\tau} \tag{4}$$

The largest time delay $\tau_{max} = 0.5$ s was iteratively decreased to suffice the coefficient of determination $R^2 \geq 0.9$ for avoiding the influence of sub-diffusion or insufficient amount of data. This followed the general approach of a Brownian particle in two-dimensions. The diffusions less than $D_{min} = 0.01$ μm$^2$ s$^{-1}$ are defined as non-diffusive considering the spatiotemporal limitations of the experimental setup and resolution of the fitted two-dimensional Gaussian function, since most led to $R^2 << 0.9$. The calculation of the mean $\mu$ and standard deviation $\sigma$ of the diffusion constant was done using the log-transformation due to its skewed normal distribution, as:

$$\mu_{log} = \frac{1}{N}\sum_{i=1}^{N} \log(D_i) \tag{5}$$

and

$$\sigma_{log} = \sqrt{\frac{1}{N}\sum_{i=1}^{N}(\log(D_i) - \mu_{log})^2} \tag{6}$$

where $N$ is the number of diffusion constants obtained for one experimental condition. The back-transformation was calculated then as

$$\mu = \exp(\mu_{log} + 0.5 \cdot \sigma_{log}^2) \tag{7}$$

and

$$\sigma = \sqrt{\mu^2 \left(\exp(\sigma_{log}^2) - 1\right)} \tag{8}$$

respectively, following the Finney estimator approach[58]. The standard error of mean (SEM) considering a confidence interval of 95% was calculated as

$$SEM = \frac{1.96\sigma}{\sqrt{N}} \tag{9}$$

**Reporting summary**. Further information on research design is available in the Nature Research Reporting Summary linked to this article.

## Data availability

All data generated or analyzed during this study are included in the paper and supporting files. All original TIRF source data are stored at the data repository of the University of Stuttgart (DaRUS): S.W., L.F., S.L., M.I.W., M.H., S.N., 2021. Data for: Correlation of mitochondrial TOM core complex stop-and-go and open-closed channel dynamics, https://doi.org/10.18419/darus-2158.

## Code availability

The Matlab software code used for data analysis can be provided upon reasonable request.

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

## Acknowledgements

We thank Beate Nitschke for help with protein preparation, Stephan Eisler, and Ke Zhou for help with the TIRF microscopy, and Sisi Fan for help with the AFM. We thank Robin Ghosh, for stimulating discussion, and the Baden-Württemberg Foundation for funding (BiofMO-6, S.N.).

## Author contributions

S.N. initiated and directed the study. S.W. fluorescently labeled proteins, collected and processed the TIRF data. M.H. and S.W. wrote the software used for data and statistical analysis. S.W., M.H., and S.N. analyzed results. S.N. wrote the initial paper draft and secured funding. S.W., M.W., M.H., and S.N. edited and reviewed the draft. L.F., S.L., and M.W. provided initial expertise.

## Funding

## Competing interests

The authors declare no competing interests.
