## [Peer Review File · Communications Biology]

Reviewers' comments:

Reviewer #1 (Remarks to the Author):

In this work, Wang et al present a new methodology to observe single pore opening in biological membranes. The method is electrode-free, relies on microscopy observations and is performed on artificial membranes supported on agarose. The authors study the opening of the mitochondrial TOM-CC complex and conclude that "TOM-CC reversibly switches between three states" and that this switching depends on molecular mobility. My main criticism is that the work does not delve into the significance of these observations and that many of the claims are not fully supported by the data presented.

Proposing that TOM-CC has a diffusion-dependent pore activity is too speculative, as the work shows no evidence of it happening in vivo or in a support (hydrogel)-free system. Such an activity would be unprecedented for a transmembrane protein and in my opinion requires a more detailed study.

As the authors point several times ("We explain this behavior by the mechanical binding of the two protruding Tom22 subunits to the hydrogel", "ostensibly due to interaction between the extended polar domains of Tom22 and the supporting agarose film" or "conformational changes of the TOM-CC are only caused by thermal motion of the protein and possible interactions with the hydrogel underlying the membrane."), the observations might derive from artefactual interactions of Tom22 with the agarose, which would greatly reduce their significance. I think it is key that the authors demonstrate whether this occurs either in vivo or in a system less prone to artefacts.

Still, the method is simple, clever and elegant, and I think it has the potential to yield new insights into the function of membrane pores and channels. However, I believe that TOM-CC might not be the best candidate to study with this new methodology.

Specific comments:

1. Page 1-36. The opening and closure of TOM-CC sometimes is completely independent of its movement, see Movie 1 for example. On some occasions the protein stops but the signal remains constant, while on others it changes while moving. This is not to say that sometimes there is not a clear correlation between calcium signal and diffusion.

However, claims such as "Freely diffusing TOM-CC molecules are observed in a high permeability state, while non-moving molecules are in an intermediate and a low permeability state." do not reflect the obtained results. This should be rephrased.

2. Page 1-40 "TOM-CC could thus be the first b-barrel protein channel to exhibit membrane state-dependent mechanosensitive properties".

At its current state this is too strong of a claim for the abstract. In my opinion, it misleads readers into believing that the Tom40 pore closes depending on mechanical phenomena, while the most likely scenario seems that it is Tom22-mediated and diffusion changes just a consequence of this. The work shows that the b-barrel protein (Tom40) does not show any mechanosensitive conformational change by itself (Figs. 2-3). I suggest omitting or rephrasing it to more accurately reflect the results presented in the paper.

3. Page 2-52. "...depending on the activity of mitochondria, the mode of lateral mobility of TOM within the mitochondrial outer membrane may be fundamental to the different import needs of the organelle".

The diffusion of TOM might change because of its interactions with other proteins, as it is the case for most proteins. However, I see no reason or precedent in the literature to claim that the diffusion mode of TOM might be fundamental to its activity.

4. Page 2-65. The authors explain that previously observed voltage-dependent conformational changes might not be "physiologically significant". Could the authors be more specific about the voltages they are referring to? What are the voltages that induced pore closure in previous works and what are the possible Donnan potentials they refer to? This would give the reader an idea of how physiologically significant these observations are.

Moreover, I think that the ion concentrations used by the authors (1.32 M KCl inside the DIB and 0.66 M CaCl₂ on the agarose side) are not physiologically significant either. The manuscript would

benefit from openly discussing this.

5. Fig. 1A. How can the authors be sure about the presence of all these proteins in the complex? It would be better to include a western blot or mass spectrometry analysis of the purified protein.

6. Page 3-12. "This enables the spatiotemporal tracking of individual molecules with much higher accuracy". I do not agree with this. The images (e.g. Fig. 2A) show bright spots in the range of several micrometres. What is the localisation precision for the protein? It is hard to believe that it is higher than in classical single particle tracking experiments (e.g. Supplementary Movie 4), with just a few nanometres, that enable accurately determining diffusion coefficients.

Could the authors discuss how this affects the diffusion coefficients calculated?

7. Is calcium an accurate reporter of the activity of the TOM-CC? I.e., does the conductance of a small cation reflect the capacity of the complex to translocate kDa-sized proteins? Since there is Fluo-8 signal also when the channels are closed, this would mean that the closed conformation is leaky to Calcium. Could the authors discuss this?

8. Page 3-32. "These data allow us to conclude that the fluctuations between the three defined permeability states are an inherent property of TOM-CC." I do not think the data fully support this claim. It can also be explained by the interaction of Tom22 with the hydrogel (as the authors suggest), which is not an inherent property of TOM-CC.

In this line, Page 3-38, I do not think the experiments with Tom40 or OmpF can discard "nonspecific interaction of the TOM-CC with the glass slide" or the agarose. Since they are mostly embedded in the membrane, as compared to TOM-CC, which might prevent such interactions with the hydrogel.

9. Page 3-49. "...the functions of many integral membrane proteins depend on their local position and state of movement in the membrane."

To the best of my knowledge, there is no evidence of the function of an integral membrane protein being coupled to its diffusion. At most, diffusion indirectly changes due to interactions with other partners (increase of size induces a decrease in diffusion coefficient). Enzymes of course need to move to carry out their activity, such as the rotation of the ATP-synthase, but this does not mean that the diffusion of a whole complex directly determines its function. Could the authors be more specific here? Examples would help. Also, what do the authors mean by state of movement?

10. Page 4-69 and Supplementary Movie 4. The fact that the proteins stop at the exact same position points towards an agarose-induced closure, as the authors note.

11. Page 4-75 "None of these protein channels show coupling between channel activity and lateral protein diffusion." and page 5-20 "Purified Tom40 itself showed no correlation between channel activity and translational diffusion in the membrane".

This is picky, but in my view, they are fully coupled/correlated; they always move and are always open. A pore-forming protein that transiently stops might be a better control.

12. Fig. 5. Throughout the manuscript the idea that molecules that stop show a closed conformation is conveyed. Fig. 5b however, shows that immobilised molecules mostly (ca. 95%) exist in the S-I state, i.e. partially open. Moreover, from this plot, it seems at least equally likely that immobilised TOM-CC exist in the S-H and S-L states, regardless of the presence of Ni.

13. The conclusion that Tom22 is the responsible for closure is reasonable, given the structure of the complex. However, there is no evidence showing that any of the other TOM-CC proteins (Tom5, 6 or 7) might not play a role. Is there previous evidence that Tom22 could control the opening of the pores? Tom22 mutants to prevent interaction with the gel would be a good control here.

14. Page 9-29. What is the reason to choose DPhPC as a lipid to model the outer mitochondrial membrane?

15. Page 9-40. Where is photobleaching in the system coming from? Is fluo-8 excited regardless of the calcium concentration?

If I understood correctly, photobleaching is corrected by using the background signal as a reference. Is the background signal from the camera accounted for? How is the "background intensity of the ROI" calculated?

Are the movies photobleaching corrected?

Also, the movement of calcium to the trans side of the membrane would cause an increase in Fluo-8 fluorescence over time. Is this observed or is bleaching dominating?

Reviewer #2 (Remarks to the Author):

The manuscript by Nussberger et al. described a study that used droplet interface membranes

(DIBs) to observe the channel activities of the TOM-CC complex, which is the main gate for the entry of nuclear-coded proteins into mitochondria. The work presented an interesting discovery that the "closure" of a Tom40 channel within a TOM-CC complex occurred when Tom22 was immobilized on the hydrogel. The manuscript was well organized and a pleasure to read. The whole data set, including results of well-designed experiments and the supplemental movies provided solid and visual evidence to support the conclusion. Although this manuscript did not offer any mechanistic interpretation for the synchronized "stop and close" behavior of TOM-CC, their findings pointed out an exciting direction to examine the role of Tom22 in regulating the channel activity of Tom-CC. Their finding is novel and would be of great interest for the community of mitochondria biogenesis and protein homeostasis. This reviewer recommends the manuscript for publication in Communications Biology with minor revision.

Some minor issues are listed here:

1. Page 5: This section was a little hard to understand. In previous sections, events with high SH were always associated with the high mobility DH. In this section, the authors discussed rare events exhibiting a high SH but a non-diffusive $DH < D_{min}$, i.e. DH not necessarily indicates high mobility here. The characteristics of these non-canonic events were not clearly explained.
2. Page 5: Also the DH, I requires a clear definition. Does DH, I indicate the average diffusion coefficients of DH and DI of a single TOM-CC complex at SH and SI state? Or it stands for "DH and DI"?
3. Page 5: The classification of three types of events was confusing. It would be helpful if the authors include a representative trace for each class of events.
4. In the "Methods", both mg ml⁻¹ and mg/ml were used in the manuscript. Please chose one form.
5. Page 8 Line 75, unfinished sentence "labeled with the fluorescent dye Cy3 according to...42"

Reviewer #3 (Remarks to the Author):

In this manuscript, authors provide robust evidence supporting that the TOM core complex (TOM-CC), a membrane β -barrel protein, has its function impacted not just by biochemical, but also mechanical stimuli. Authors developed a simplified in vitro system to observe TOM-CC activity and diffusion, which does not fully mimic physiological contexts, but allows the identification of a correlation between TOM-CC movement and cis-trans influx. These are very interesting findings, especially because the physiological role of mitochondrial function in cell homeostasis has been extensively investigated in different cell types, strengthening a causal link between mitochondrial metabolism and cell fate and function.

There are, however, some points that would need to be addressed for further clarification of authors conclusions.

1. Page 3 Line 19: Authors raise the possibility that "high and intermediate intensity levels correspond to two conformational states", which would reflect pore opening. This idea is sustained along the manuscript. Is there a way to address that experimentally? Or this is justified by the 3 observed bands of fluorescence intensity? Page 6 line 94, authors say that "Functional reversible disassembly and reassembly of the TOM-CC at time scales studied here are highly improbable." How can authors exclude the presence of "single barrels"? Finally, if both pores are closed, what justifies the leakage of fluorescence observed in the low intensity level conformation?

2. The use of Ni-NTA modified agarose indeed suggests there is a causal link between trapping and immobility, which leads to higher frequencies of SI and SH conformations (Fig. 4 and Fig 5). However, what could be the molecular trap or anchor point that would trigger a short-interaction between TOM-CC (Tom22) and anything below the membrane in the supporting hydrogel (first mentioned in Page 4 Line 71)? In movie S3, for example, it is clear that a SH TOM-CC moves very close to a SI TOM-CC. This could mean a very specific interaction/stopping point below the membrane is being occupied. Can authors exclude that the lipid bilayer itself or the matrigel architecture could be causing this phenomenon? Authors mention in Page 5 line 39 that "Another possible but unlikely explanation is a spatial void of agarose network". Could authors definitely exclude that the hydrogel architecture could be having an impact by imaging it?

3. Could authors explore whether any loss-of-function Tom22 mutants could have their functional impairment explained by changes in their mechano-sensitivity?

4. Signal to noise ratio in video S7 is much lower for the TOM-CC channels tracked on Ni-NTA modified agarose. Could this suggest that permeability is generally compromised by Tom-22 interaction with the hydrogel network? Was this consistently the case when using Ni-NTA modified agarose?

5. I believe authors should further discuss what could be the physiological relevance and implications of TOM-CC's mechano-sensitivity. Previous reports, also referred to by the authors, show that TOM-CC can interact with different proteins in mitochondrial outer- and inner-membranes, and these different interactions lead to different functional outcomes. Based on the author's results, TOM-CC is in an open conformation when moving: how would this work in a real mitochondrion, where TOM-CC facilitates influx into the inner membrane or matrix by interacting with other proteins? Do authors suggest these interactions then happen with other proteins that are also highly diffusive?

Minor points:

1. In Fig.1: authors might consider changing the colour of Ca²⁺ (now red and described as red in the Fig. legend) as the red is also used in the lipid membrane.

2. Authors never mention what is a.u. in the y axis of several graphs. I assume it is arbitrary unit, but it should appear at least in the figure legend.

3. In video S5 there seems to be a TOM-CC complex that presents lower fluorescence intensity (compatible to SI) than the highlighted ones that is also moving.

Reviewer #1:

In this work, Wang et al present a new methodology to observe single pore opening in biological membranes. The method is electrode-free, relies on microscopy observations and is performed on artificial membranes supported on agarose. The authors study the opening of the mitochondrial TOM-CC complex and conclude that “TOM-CC reversibly switches between three states” and that this switching depends on molecular mobility. My main criticism is that the work does not delve into the significance of these observations and that many of the claims are not fully supported by the data presented. Proposing that TOM-CC has a diffusion-dependent pore activity is too speculative, as the work shows no evidence of it happening in vivo or in a support (hydrogel)-free system. Such an activity would be unprecedented for a transmembrane protein and in my opinion requires a more detailed study.

We thank the reviewer for her/his detailed comments to further clarify our conclusions and improve our manuscript. We have considered all recommendations point by point in the following response.

We fully agree with our reviewer that the diffusion-dependent pore activity is so far unprecedented for a transmembrane protein, although there is growing evidence that the surface membrane dynamics of voltage-gated ion channels are also relevant to function in their physiological context (for review, see Heine et al. 2016 Channels 10:267-281). The same could also apply to TOM as described in this paper.

Kuzmenko et al. (Sci. Rep 1:195, 2011) and Appelhans et al. (Biophys. Rev. 9:345-52, 2017) have shown that Tom40 exhibits a highly dynamic but confined diffusion behavior in isolated mitochondria. This would be entirely consistent with the stop-and-go behavior of TOM-CC observed in our work. In contrast to the in vitro situation, the in vivo time window for stop and go may be much smaller, thereby more demanding for experimental study. This may be the reason that the open-closed behavior in vivo has not yet been considered explicitly.

- High-resolution imaging techniques have shown that many ion channels are not static, but are subject to highly dynamic processes that include transient association of pore-forming and auxiliary subunits, lateral diffusion, and clustering with other channels, which in turn affects their function (for review, see Heine et al., 2016, Channels 10 267-281; Jacobson et al. 2019, Cell 4 806-819). We therefore think that the observed correlation between the stop-and-go and open-closed behavior of TOM-CC provides an intriguing new conceptual idea we would like to share with the scientific community. Nevertheless, we fully agree that it will ultimately be necessary to demonstrate a similar behavior for the TOM-CC in vivo.
- Establishing an in vitro assay that allows studying the mechanism of bacterial or eukaryotic protein translocases at the single molecule level is overdue. Such an assay is required to provide a biophysical answer to the fundamental and long-standing question of how unfolded or partially folded polypeptide chains undergoing thermal fluctuations can partition and subsequently transfer through a nanometer scale pore. In this context, the assay using mitochondrial TOM-CC and simultaneous single-molecule TIRF imaging of both channel activity and lateral movement of the complex in lipid membranes, as presented in the current manuscript, reveals a new and unexpected property of the TOM-CC. At this step we are not attempting to prove the in vivo situation, but describe the setup for a bottom-up approach. To further support our in vitro conclusions, we present additional new data on OmpF used as a control (see response to comment 8). OmpF exhibits similar stop-and-go behavior to TOM-CC, albeit much less frequently than TOM-CC. However, its channel activity does not seem to be influenced by the type (“stop” vs. “go”) of movement.
- Regarding the proposal to investigate the correlation between the stop-and-go and the open-closed behavior of the TOM-CC in a support (hydrogel)-free system, we would like to note that in our type of biophysical approach a supported lipid bilayer is mandatory to achieve optical single molecule resolution. In fact, this might even mimic transient in vivo interactions of TOM-CC (Tom22) with components of TIM23 complex (Gomkale et al. 2021, Nat. Com. 12: 5715) in the periphery of the mitochondrial outer membrane.

To clarify this point in the revised manuscript, we have added a few sentences in the Discussion section that also follow the suggestion of the Reviewer #3 to “further discuss the physiological relevance and implications of TOM-CC’s mechanosensitivity”.

Page 8, lines 345-355, now reads: “With regard to the mechanism of TOM-mediated protein translocation into mitochondria, our data open the exciting possibility that proteins at the periphery

of the outer membrane, for example components of inner membrane TIM23 complex (Gomkale et al. 2021, Nat. Com. 12: 5715), might not only limit the diffusion of the TOM-CC by transient interaction and formation of mitochondrial contact sites, but also regulate the open-closed activity status of the channel. This could in turn translate into conformational changes of the TOM-CC that influence its translocation process of incoming mitochondrial preproteins. In this context, it might be interesting to visualize different Tom22-dependent conformations of TOM-CC using cryo-EM and to perform single-molecule FRET experiments to study the dynamics of TOM-CC between different conformations in the near future.”

As the authors point several times (“We explain this behavior by the mechanical binding of the two protruding Tom22 subunits to the hydrogel”, “ostensibly due to interaction between the extended polar domains of Tom22 and the supporting agarose film” or “conformational changes of the TOM-CC are only caused by thermal motion of the protein and possible interactions with the hydrogel underlying the membrane.”), the observations might derive from artefactual interactions of Tom22 with the agarose, which would greatly reduce their significance. I think it is key that the authors demonstrate whether this occurs either in vivo or in a system less prone to artefacts.

Our study indicates that the stop-and go behavior of the TOM-CC correlates with channel activity. We fully agree that the simplest molecular mechanism would be a conformational change of the complex, due to an interaction the TOM-CC with the agarose layer. With the relative distances involved between the membrane and the agarose layer only the two Tom22 subunits, which extend well into the aqueous layer, are readily available for mediating this conformational change. Whether this is “artefactual” in vivo or not remains to be seen. However this “artefactual” observation does not mitigate the result of the study. We therefore stand by the in vitro conclusions presented in this manuscript.

Still, the method is simple, clever and elegant, and I think it has the potential to yield new insights into the function of membrane pores and channels. However, I believe that TOM-CC might not be the best candidate to study with this new methodology.

The use of our methodology to study the dynamics of membrane protein complexes is indeed challenging. Nevertheless, we believe that the approach is also worthwhile for observing the molecular channel dynamics of TOM, as it offers many advantages over approaches that use classical electrophysiology. Here we can “simultaneously” study both channel activity and lateral membrane protein diffusion in membranes. In particular, we can observe the thermal motion of the TOM-CC channel without the interference of external electrical forces or possible diffusional artefacts due to the fusion of engineered proteins with fluorescent protein tags.

Specific comments:

1. Page 1-36. The opening and closure of TOM-CC sometimes is completely independent of its movement, see Movie 1 for example. On some occasions the protein stops but the signal remains constant, while on others it changes while moving. This is not to say that sometimes there is not a clear correlation between calcium signal and diffusion.

We agree with the reviewer that one of the TOM-CC molecules in Movie 1 briefly stops its motion at $t = 6.99$ s for about one second and the intensity drops by only 13.4 %, which is not quite the S_I state (Movie S2 and Fig. 2b). However, 50 ms before resuming motion and fully opening to S_H at $t = 7.94$ s, it briefly enters the S_I state with intermediate intensity, as expected. A few milliseconds later it stops again and enters the fully dark S_L state. Unfortunately, such observations cannot be completely ruled out in single-molecule experiments and can only be evaluated by statistical analyses.

To improve our manuscript, we have now highlighted this in the legend of Supplementary Movie S1 and refer to the statistics shown in Fig. 5b.

We indicate in the legend: “The intensity trace of the upper right spot is shown in Movie S2 and in Fig.2b. The probabilities for TOM-CC being in states S_H , S_I and S_L are presented in Fig.5.”

However, claims such as “Freely diffusing TOM-CC molecules are observed in a high permeability state, while non-moving molecules are in an intermediate and a low permeability state.” do not reflect the obtained results. This should be rephrased.

We thank the reviewer for pointing out that we should be more careful in our formulation, as not all freely diffusing TOM-CC molecules are in a high permeability state, while non-moving molecules are in a state of intermediate or low permeability. This is not surprising as individual membrane proteins can undergo spontaneous conformational changes due to thermal motion. At the suggestion of the reviewer, we have

therefore re-worded our conclusion in the abstract (page 1, lines 37-38), which now reads: "While freely diffusing TOM-CC molecules are generally in a high permeability state, non-mobile molecules are mostly in an intermediate or low permeability state."

2. Page 1-40 "TOM-CC could thus be the first β -barrel protein channel to exhibit membrane state-dependent mechanosensitive properties". At its current state this is too strong of a claim for the abstract. In my opinion, it misleads readers into believing that the Tom40 pore closes depending on mechanical phenomena, while the most likely scenario seems that it is Tom22-mediated and diffusion changes just a consequence of this. The work shows that the β -barrel protein (Tom40) does not show any mechanosensitive conformational change by itself (Figs. 2-3). I suggest omitting or rephrasing it to more accurately reflect the results presented in the paper.

The referee is correct in pointing out that the Tom40 β -barrel itself does not appear to exhibit mechanosensitive conformational changes, but that the Tom40 pore closures are likely mediated by the two Tom22 subunits.

We have now clarified this in the abstract (Page 1, lines 39-43), stating that "the behavior of the TOM-CC by the mechanical binding of the two protruding Tom22 subunits to the hydrogel and a concomitant combinatorial opening and closing of the two β -barrel pores of TOM-CC. TOM-CC could thus be the first β -barrel membrane protein complex to exhibit membrane state-dependent mechanosensitive properties, mediated by the two Tom22 subunits".

3. Page 2-52. "...depending on the activity of mitochondria, the mode of lateral mobility of TOM within the mitochondrial outer membrane may be fundamental to the different import needs of the organelle". The diffusion of TOM might change because of its interactions with other proteins, as it is the case for most proteins. However, I see no reason or precedent in the literature to claim that the diffusion mode of TOM might be fundamental to its activity.

From our point of view it seems that this aspect has not really been considered in the literature. However, it is well known that deformation of a membrane bilayer as well as external interactions with peripheral proteins can have significant effects on membrane protein diffusion and function (for review, see Jacobson et al. 2019, Cell 177:806-819). Differential lateral membrane diffusion (trapped versus moving) was also observed for the TOM-CC subunits Tom40 and Tom7 in immobilized isolated mitochondria (Kuzmenko et al. 2011 Sci. Rep 1:195; Appelhans et al. 2017 Biophysical Reviews 9:345-352; Wilkens et al. 2013 J. Cell Sci. 126:103-116, Bhagawati et al. 2021 Mol. Biol. Cell 32:1-11). However, a possible correlation with channel activity was not investigated. Whereas we cannot unambiguously confirm this for the in vivo situation, our studies show the correlation to be true for an in vitro system. We therefore believe that our conclusion is warranted for the in vitro system.

4. Page 2-65. The authors explain that previously observed voltage-dependent conformational changes might not be "physiologically significant". Could the authors be more specific about the voltages they are referring to? What are the voltages that induced pore closure in previous works and what are the possible Donnan potentials they refer to? This would give the reader an idea of how physiologically significant these observations are.

At the suggestion of the reviewer, we have re-worded this paragraph to why electrophysiological experiments on TOM-CC do not necessarily describe the real dynamics of its two protein channels. In the Introduction section (page 2, lines 68-74), it is now noted "that the physiological significance of the voltage-dependent conformational transitions between the open and closed states of the TOM-CC is controversial because the critical voltage $|\Delta V| > 50$ mV above which TOM-CC channels close (Künkele et al. 1998 Cell 93:1009-19; Poynor et al. 2008 Biophys. J. 95:1511-22), is significantly greater than any metabolically theory-derived potential $|MDP| < 5$ mV at the outer mitochondrial membrane (Lemeshko et al. 2000 Biophys. J. 79:2785-2800)."

Moreover, I think that the ion concentrations used by the authors (1.32 M KCl inside the DIB and 0.66 M CaCl₂ on the agarose side) are not physiologically significant either. The manuscript would benefit from openly discussing this.

At the suggestion of the reviewer, we have now taken up this point in the legend to Fig. 1 (page 15, lines 736-739) and openly state: "The cis side of the membrane contained Ca²⁺-ions (0.66 M), while having at the trans side a Ca²⁺-sensitive fluorescent dye (Fluo-8) and KCl (1.32 M) to balance the membrane osmotic pressure. The high Ca²⁺ content is necessary to induce high calcium flux through the TOM-CC pores and to provide reliable optical signals."

The ionic conditions used follow the recommendations described by Mark Wallace (now co-author) and Hagan Bayley for optical monitoring ion transfer through single ion channels (Villar et al. 2011 Nature Nanotech. 6:803–80; Lepthin et al. 2013 Nature Protoc. 8:1048–57). At a fundamental level, a high concentration of KCl is necessary to neutralize the high ionic strength at the membrane surface at the microscopic level. This obviates possible artefacts due to calcium binding to the surface of the phospholipids. High salt levels have been shown previously to facilitate liquid crystalline motion of phospholipids by NMR spectroscopy. In this context the rapid movement of the lipid in the bilayer can be considered to be physiological. High salt levels are also often used for classical electrophysiological experiments. The high levels of calcium are necessary here for inducing high calcium flux through the TOM-CC pores and to provide reliable optical signals. The most common artefact introduced by calcium, protein aggregation, is not observed.

5. Fig. 1A. How can the authors be sure about the presence of all these proteins in the complex? It would be better to include a western blot or mass spectrometry analysis of the purified protein.

We have demonstrated in previous work by blue native gel electrophoresis, size exclusion chromatography, Western blot, native mass spectrometry and high-resolution MALDI-MS (Ahting et al. 1999 J. Cell. Biol. 147:959-68; Mager et al. 2020 J. Phys. Condens. Matter 22:454132 7pp, Bausewein et al. 2017 Cell 170: 693-700) that all Tom proteins are present in isolated TOM-CC. We were also the first group to show by single molecule electron tomography that the TOM-CC is a two-pore complex (Ahting et al. 1999 J. Cell. Biol. 147:959-68), which was later confirmed with all its 10 subunits by high-resolution cryoEM in collaboration with Werner Kühlbrandt (Bausewein et al. 2017 Cell 170: 693-700). Since the TOM-CC used in this study was prepared identically to the previous work we prefer to cite the original literature where our preparation has been described in detail.

To address the question as to which Tom proteins were present in the complex, additional references to the composition and stoichiometry of all subunits of isolated TOM-CC are now provided not only in the Materials and Methods (page 9, line 397), but also in the Results section (page 3, line 103).

6. Page 3-12. “This enables a the spatiotemporal tracking of individual molecules with much higher accuracy”. I do not agree with this. The images (e.g. Fig. 2A) show bright spots in the range of several micrometres. What is the localisation precision for the protein? It is hard to believe that it is higher than in classical single particle tracking experiments (e.g. Supplementary Movie 4), with just a few nanometres, that enable accurately determining diffusion coefficients.

We apologize for the confusion and are grateful this point has been spotted. The confusion is caused by a mistake on our part in using the term “spatiotemporal” when we should have used “temporal”. We wanted to point out that our experiments allowed the recording of diffusion trajectories of TOM-CC up to several minutes at high temporal resolution without the use of interfering fluorescent labels that could alter its diffusion behavior beyond a general reduction of its native diffusion rate. In order to accurately resolve and characterize the different diffusion states of the TOM-CC, sufficiently long trajectories and well-sampled distributions of experimental measurements of fluorescent molecules are required. The longer the time series, the better. The localization precision for the protein is given by the accuracy of fitting the spots to the two dimensional Gaussian function and thus ultimately by the pixel size (for reference, see Materials and Methods).

We clarified this in the text and deleted the original sentence.

Could the authors discuss how this affects the diffusion coefficients calculated?

As pointed out in Materials and Methods (page 11, lines 510-522) the lateral diffusion constants were obtained from individually for spots within the respective high and intermediate amplitude range by linear regression of the time delay τ and the mean square displacement of the spots. The largest time delay $\tau_{\max} = 0.5$ s was iteratively decreased to suffice the coefficient of determination $R^2 \geq 0.9$ for avoiding the influence of sub-diffusion or insufficient amount of data. In single particle tracking experiments using single fluorescent molecules, blinking and photobleaching can reduce the amount of data, potentially reducing R^2 if the sampling rate is not high enough.

7. Is calcium an accurate reporter of the activity of the TOM-CC? I.e., does the conductance of a small cation reflect the capacity of the complex to translocate kDa-sized proteins? Since there is Fluo-8 signal also when the channels are closed, this would mean that the closed conformation is leaky to Calcium. Could the authors discuss this?

Clearly calcium is not a kDa-size protein. Nevertheless, calcium diffusion through channels is a useful tool to proving the open-closed dynamics of channels in general. The logic therein lies in the assumption

that the lack of calcium penetration can only be due to channel closing. An additional consideration is that the hydration diameter of calcium is 0.82 nm (Israelachvili 2011, in: Intramolecular and Surface Forces, Academic Press, 3rd Edition), which is not so different from the effective diameter of an extended peptide. However, there are ambiguities associated with the method, but in general these do not detract from the goal of measuring open-closed activity. It is not surprising that channels can also be leaky due to thermal activation.

We have now added this to the revised version of the manuscript (page 3, lines 123-124) and state that "the low intensity level may represent a conformation (S_L) where both pores are closed with residual permeation for calcium".

8. Page 3-32. "These data allow us to conclude that the fluctuations between the three defined permeability states are an inherent property of TOM-CC." I do not think the data fully support this claim. It can also be explained by the interaction of Tom22 with the hydrogel (as the authors suggest), which is not an inherent property of TOM-CC.

As indicated by high-resolution cryo-EM (Bausewein et al. 2017 Cell 170: 693–700) and native LILBID mass spectrometry (Mager et al. 2010 J. Phys. Condens. Matter 22: 454132), two Tom22 subunits are intrinsic subunits of TOM-CC. They connect the two Tom40 β -barrel transmembrane channels. The interaction of Tom22 with the hydrogel and the associated fluctuations between the three defined permeability states of the TOM-CC are thus an inherent property of the TOM-CC. Following the definition of the term "inherent", we wanted to say that the fluctuations between the three defined permeability states a characteristic feature of the TOM-CC.

Nevertheless, to avoid semantic confusion we have deleted this sentence and leave the interpretation of our data to the scientific community.

In this line, Page 3-38, I do not think the experiments with Tom40 or OmpF can discard "nonspecific interaction of the TOM-CC with the glass slide" or the agarose. Since they are mostly embedded in the membrane, as compared to TOM-CC, which might prevent such interactions with the hydrogel.

We thank the reviewer for this comment. To better elucidate the differences and possible similarities of our control protein OmpF with TOM-CC, we recorded another 42 single-molecule TIRF-M videos of OmpF in DIB membranes and examined a total of 171 OmpF molecules for possible stop-and-go behavior.

- As shown in our initial submission, most OmpF molecules exhibit the most basic type of mobility expected for homogeneous membranes: simple Brownian translational diffusion (73 %, $n_{\text{OmpF}} = 125$).
- Due to the higher number of images analyzed ($N = 487,008$) and the higher number of OmpF molecules examined, we now also see permanently trapped, non-moving OmpF molecules (12 %, $n_{\text{OmpF}} = 21$). They have practically the same intensity as moving OmpF molecules.
- Some OmpF molecules (15 %, $n_{\text{OmpF}} = 25$) show a stop-and-go behavior similar to that of TOM-CC, indicating a transient interaction with the hydrogel under the membrane. Unlike TOM-CC, the intensities of OmpF do not change when switching from "go" to "stop" and vice versa. They are relevant as negative controls.

We have now included this new data in the revised version of our manuscript and rephrased the Results section on page 3 (lines 136-146) and on page 4 (lines 178-186). The new trajectories and corresponding intensity traces are now shown in two additional figures in the supplement (Supplementary Figures S6 and S7). The new paragraphs now read as follows:

Page 3, lines 136-146: "To rule out the possibility that the observed intensity fluctuations are caused by possible thermodynamic undulations of the membrane in the evanescent TIRF illumination field or by local variations in Ca^{2+} flux from cis to trans, we compared the ion flux through TOM-CC with that through isolated individual Tom40 molecules and an unrelated multimeric β -barrel protein which has three pores and is almost entirely embedded in the lipid bilayer. In a series of control experiments, we reconstituted Tom40 (Figs.2e-f and Supplementary Fig.S4) and E. coli OmpF (Figs.2e-f and Supplementary Figs.S5-S7) into DIB membranes and observed virtually constant fluorescence intensities, respectively. In contrast to TOM-CC, neither protein channel exhibits gating transitions between specific permeability states. Since Tom40 consists of a monomeric β -barrel channel (Ahting et al. 2001, 153, 1151-1160), the

toggle of the TOM-CC between the three different permeability states S_H , S_I and S_L (Fig. 2c) must therefore have another root cause.”

Page 4, lines 178-186: “In contrast, most single Tom40 molecules isolated from TOM-CC (Supplementary Movie S5, Figs.3e and Supplementary Fig.S4) and OmpF molecules purified from *E. coli*. outer membranes (Supplementary Movie S6 and Supplementary Fig.S5) show the most elementary mode of mobility expected for homogeneous membranes: simple Brownian translational diffusion. Consistent with the fact that OmpF is fully embedded in membranes, only very few out of a total of 171 analyzed OmpF molecules were temporarily (~15%) or permanently (~12%) trapped in the membrane (Supplementary Figs. S7 and S6). In contrast to TOM-CC, however, a “stop” of OmpF did not seem to be accompanied by a change in intensity (Supplementary Figs. S6 and S7) and thus by closing the pores.”

9. Page 3-49. “... the functions of many integral membrane proteins depend on their local position and state of movement in the membrane.”

To the best of my knowledge, there is no evidence of the function of an integral membrane protein being coupled to its diffusion. At most, diffusion indirectly changes due to interactions with other partners (increase of size induces a decrease in diffusion coefficient). Enzymes of course need to move to carry out their activity, such as the rotation of the ATP-synthase, but this does not mean that the diffusion of a whole complex directly determines its function. Could the authors be more specific here? Examples would help. Also, what do the authors mean by state of movement?

We thank the reviewer for this comment and have now re-worded our statement to clarify this point and added some examples. Page 3 (lines 149-159) now reads as follows:

“Lateral mobility, e.g. free, restricted or directed diffusion, is an important factor for the organization and function of biological membranes and its components (Owen et al. 2009, *Traffic* 10:962-971; Spector et al., 2010, *Biophys. J.* 99:2880-3886; Alenghat et al. 2013, *Curr. Top. Membr.* 72: 89–120, Heine et al., 2016; *Channels* 10: 267–281; Kusumi et al. 2014, *Nature Chemical Biology* volume 10: 524–532; Jacobson et al. 2019, *Cell* 4 806-819). The transient anchoring of mobile membrane proteins thereby often precedes protein-induced signaling events, as shown for CFTR and presynaptic calcium channels (Haggie et al. 2006, *Mol. Biol. Cell* 17:4937-4945; Schneider et al. 2015, *Neuron* 86:672-679). It is also well accepted that there is a direct correlation between activity and rotational membrane diffusion for cytochrome oxidase and sarcoplasmic reticulum ATPase (Kawato et al. 1981, *J. Biol. Chem.* 256: 7518-7527; Thomas et al. 1978, *PNAS* 75:5488-5492; Hoffmann et al., 1979, *PNAS* 76:3860-3864). Since rotational diffusion is obligatorily coupled to lateral diffusion (both require a fluid crystalline membrane), we asked whether, and if so, how the mode of diffusion affects the channel activity of TOM-CC and its permeability states S_H , S_I and S_L . To this end, we simultaneously tracked the open-closed activity and position of individual TOM-CC molecules in the membrane over time (Fig.3a).”

10. Page 4-69 and Supplementary Movie 4. *The fact that the proteins stop at the exact same position points towards an agarose-induced closure, as the authors note.*

We agree and have noted this in the manuscript.

11. Page 4-75 “None of these protein channels show coupling between channel activity and mode of lateral protein diffusion.” and page 5-20 “Purified Tom40 itself showed no correlation between channel activity and translational diffusion in the membrane”.

This is picky, but in my view, they are fully coupled/correlated; they always move and are always open. A pore-forming protein that transiently stops might be a better control.

We found this comment to be very insightful and looked for a pore-forming protein that stops temporarily as a control. As also described for comment 8, we have now recorded many more movies of OmpF in agarose-supported DIB membranes. Coincidentally, some OmpF molecules do not move, indicating strong binding to the hydrogel. The spot intensity (channel activity) of both transiently and permanently trapped molecules is the same as that of laterally diffusing molecules.

We have included this new data in the revised version of our manuscript on page 3 (lines 136-146) and on page 4 (lines 178-186). The new trajectories and corresponding intensity traces are now shown in new Supplementary Figures S6 and S7. The numbering of all other figures has been adjusted accordingly in the manuscript.

12. Fig. 5. Throughout the manuscript the idea that molecules that stop show a closed conformation is conveyed. Fig. 5b however, shows that immobilised molecules mostly (ca. 95%) exist in the S-I state, i.e. partially open. Moreover, from this plot, it seems at least equally likely that immobilised TOM-CC exist in the S-H and S-L states, regardless of the presence of Ni.

The reviewer realizes correctly that the state probabilities of class III particles in S_H , S_I and S_L are almost identical regardless of presence of Ni-NTA. Nevertheless, it is important to note that a total of 67% of class III particles were found in the presence of Ni-NTA, but only 28% of class III particles were found in membranes supported by unmodified agarose. From a mechanistic point of view, it makes perfect sense that the state probabilities, which are divided into classes I, II and III, are similar regardless of the presence of Ni-NTA.

13. The conclusion that Tom22 is the responsible for closure is reasonable, given the structure of the complex. However, there is no evidence showing that any of the other TOM-CC proteins (Tom5, 6 or 7) might not play a role. Is there previous evidence that Tom22 could control the opening of the pores? Tom22 mutants to prevent interaction with the gel would be a good control here.

We agree with the reviewer that it would be important to provide evidence that Tom22 controls the opening of the pores of the TOM-CC by generating Tom22 mutants to prevent interaction with the gel. However, this possibility might be difficult to realize in the near future, especially since two Tom22 molecules stabilize the TOM-CC complex. Deletion of Tom22 may simply abolish the activity of the two-pore TOM-CC.

Nevertheless, there is some evidence (now mentioned in the Discussion, Page 6, lines 314-319) that Tom22 controls pore opening, as shown by electrophysiological analysis of TOM in outer membrane vesicles isolated from *tom22Δ* yeast mitochondria (van Wilpe et al., 1999, Nature 401:485-489). The data showed conductance states of a single channel that resembled those of purified Tom40 (Hill et al. 1998 Nature 395:516–521; Ahting et al. 2001, J Cell Biol. 2001 153:1151-1160) and differed from the coupled opening of two individual channels.

The new paragraph now reads as follows: “Previous electrophysiology studies (van Wilpe et al 1999, Nature 401) have shown that the TOM channel of yeast mitochondria is mainly in a closed state, while TOM lacking Tom22 has been found mainly in an open state, similar to that of yeast and *N. crassa* Tom40 (Hill et al Nature 1998, Ahting et al JCB 2001; van Wilpe et al 1999, Nature 401). It has therefore been postulated that Tom22 negatively regulates the opening probability of TOM (van Wilpe et al 1999, Nature 401). However, the molecular mechanism of how Tom22 influences the activity of the open-closed channels of the TOM machinery has remained an open question.”

14. Page 9-29. What is the reason to choose DPhPC as a lipid to model the outer mitochondrial membrane?

DPhPC is a commonly used lipid to produce stable membrane systems (for reference, Lindsey et al., 1979 Biochim Biophys Acta 555:147-167, Su et al., Langmuir 35:8452-8459; Villar et al., 2013, Science 340:48-52; Cao et al., 2019 Nat. Comm. 10:5668; Rosen et al., 2020 Comm Bio. 3:160). It has a branched structure and no gel/phase transition temperature. DPhPC's principal advantage in artificial bilayers is the absence of oxidizable or hydrolysable bonds in its fatty acid chains.

15. Page 9-40. Where is photobleaching in the system coming from?

As indicated in the Material and Methods section, the average total background intensity of all our time series follows a bi-exponential decrease in fluorescence intensity over time, indicating two different Fluo-8 populations with initial intensities a_1 and a_3 and rates a_2 and a_4 , respectively (Vicente et al., 2007 J. Phys.: Conf. Ser. 90 012068). We hypothesize that the decrease in fluorescence over time is due to bleaching of Fluo-8 bound to calcium (population I with rate a_2) and diffusion out of the evanescent field from the membrane into the bulk phase of the droplet (Fluo-8 population II and rate a_4).

To address this point, we have included a new figure in the supplement showing the double exponential decrease in background fluorescence intensity of a typical 512 x 512 pixels image over time (Supplementary Fig. S2).

Is fluo-8 excited regardless of the calcium concentration?

Yes, in our experiments the calcium sensitive dye is excited regardless of the calcium concentration. Fluo-8 is a calcium sensitive fluorophore with a high affinity to calcium.

If I understood correctly, photobleaching is corrected by using the background signal as a reference. Is the background signal from the camera accounted for? How is the “background intensity of the ROI” calculated?

We thank the reviewer for his/her suggestion to better explain in our manuscript how the background signal was taken into account and how the intensity of the ROI was calculated. In the revised version of our manuscript (page 10, lines 480-504) we have taken this hint and now describe the standard procedure in more detail.

“The effect of bleaching was corrected by applying a standard fluorescence bleaching correction procedure (Vicente et al., 2007 J. Phys.: Conf. Ser. 90 012068). A double exponential decay obtained by least-square fitting of the image series (average frame intensity $\langle I(t)_{\text{raw}} \rangle$) was obtained as

$$f(t, a_k) = a_1 \exp(a_2 t) + a_3 \exp(a_4 t)$$

where t is the frame index (time) and a_k are the fitting parameters. The time series was corrected according to

$$I(t) = I_{\text{raw}}(t) / f(t, a_k)$$

where $I(t)$ is the intensity as a function of time t . No filter algorithm was applied. The initial spatial position of a fluorescence spot was manually selected and within a defined region of interested (ROI, 30 x 30 pixels) fitted to a two-dimensional symmetric Gaussian function with planar tilt that accounts for possible local illumination gradients, that the global bleach correction cannot account of, as follows

$$G_{2D}(\mathbf{x}, \boldsymbol{\mu}, p_k) = p_1 + p_{2,3}(\mathbf{x} - \boldsymbol{\mu}) + A \exp(-(\mathbf{x} - \boldsymbol{\mu})^2 / 2\sigma^2)$$

where $\mathbf{x} = (x, y)$ is the ROI with the fluorescence intensity information, A and σ are the amplitude and width of the Gaussian, p_k are parameters that characterize the background intensity of the ROI, and $\boldsymbol{\mu} = (x_0, y_0)$ defines the position of the Gaussian.

Are the movies photobleaching corrected?

We apologize that this point may not have been made clear enough in the legends to our video sequences. In the original version of our manuscript, we mention in the legends for movies S1, S3, S4, S5, S6, S7 and S8 that “raw image data is shown”. In this way we wanted to express that the images correspond to the original data and have not been corrected in any way, i.e. have not been subjected to any fluorescence bleach correction or filter algorithm (as evidence, some of our images still show Newtonian ring interference patterns generated by the optical system).

To further clarify this, we have now added to all legends of our video sequences, where appropriate, that the raw TIRF-images “have not been corrected by fluorescence bleaching or a filter algorithm”.

Also, the movement of calcium to the trans side of the membrane would cause an increase in Fluo-8 fluorescence over time. Is this observed or is bleaching dominating?

We agree that the movement of calcium to the trans side of the membrane can cause an increase in Fluo-8 fluorescence over time. This can indeed be observed after several hours and incorporation of many open ion channels into the DIB membrane. However, on the observation time scale of TOM-CC channel gating in the range of a few milliseconds and total recording timescales in the range of minutes, this can be neglected; in our experiments, bleaching of fluorescent Fluo-8 in the evanescent field near to the membrane seems to be predominant.

As mentioned above, to clarify this point in the revised manuscript, we have added a few sentences to the legend of Fig.2b and a new figure in the supplement (Supplementary Fig.S2) with a typical fluorescence bleaching curve calculated from 2848 image frames corresponding to a typical time series of 60 s. In new Supplementary Fig.S2 we show how our video sequences were corrected for fluorescence bleaching. The data correspond to the whole image series shown in Movie S1 and Fig.2a. It can be clearly seen that the background Fluo-8 fluorescence does not increase with time and that bleaching of Fluo-8 in the evanescent field predominates.

Reviewer #2:

The manuscript by Nussberger et al. described a study that used droplet interface membranes (DIBs) to observe the channel activities of the TOM-CC complex, which is the main gate for the entry of nuclear-

coded proteins into mitochondria. The work presented an interesting discovery that the “closure” of a Tom40 channel within a TOM-CC complex occurred when Tom22 was immobilized on the hydrogel. The manuscript was well organized and a pleasure to read. The whole data set, including results of well-designed experiments and the supplemental movies provided solid and visual evidence to support the conclusion. Although this manuscript did not offer any mechanistic interpretation for the synchronized “stop and close” behavior of TOM-CC, their findings pointed out an exciting direction to examine the role of Tom22 in regulating the channel activity of Tom-CC. Their finding is novel and would be of great interest for the community of mitochondria biogenesis and protein homeostasis. This reviewer recommends the manuscript for publication in Communications Biology with minor revision.

We thank the reviewer for the positive feedback and have incorporated all suggestions into the revised version of our manuscript to further improve our work.

Some minor issues are listed here:

1. Page 5: This section was a little hard to understand. In previous sections, events with high SH were always associated with the high mobility DH. In this section, the authors discussed rare events exhibiting a high SH but a non-diffusive DH < Dmin, i.e. DH not necessarily indicates high mobility here. The characteristics of these non-canonic events were not clearly explained.

At the suggestion of the reviewer, we re-wrote the results section "Correlation between lateral motion and TOM-CC channel activity" (page 5, lines 238–289) to make the statistical analysis of our results and the characteristics of the non-canonical events clearer.

Moving TOM-CC molecules in the fully open SH state are now described by a diffusion coefficient $D(S_H) > D_{min}$; rare events of non-diffusing molecules in SH are described by diffusion coefficients $D(S_H) \leq D_{min}$. Moreover, the three classes of events are now described using the notation of classical set theory.

- The first and most important class I of TOM-CC molecules $\{I|D[S_H] > D_{min} \cap D[S_I] \leq D_{min}\}$ shows high lateral mobility in SH while bound and immobile in SI.
- The second class II of molecules $\{II|D[S_H] > D_{min} \cap D[S_I] > D_{min}\}$ shows high mobilities in both states, SH and SI. It is unlikely that the TOM-CC molecules of this class have a functional Tom22 or are incorporated into the membrane in the reverse orientation and therefore do not interact with the hydrogel. Another possible but unlikely explanation is a spatial void in the agarose network that prevents Tom22 from interacting with the network within the observation time window. The rare partial closure of the TOM-CC channels may occur spontaneously due to thermal motion.
- The third class III $\{III|D[S_I] \leq D_{min} \cap D[S_H] \leq D_{min}\}$ represents events with permanently bound TOM-CC molecules that do not move in the membrane and are exclusively non-diffusive (Supplemental Fig. S9a-b). Most of the molecules in this class are in SI and SL. The TOM-CC molecules that briefly move from SI to SH and back to SI (Supplemental Fig. S9c-d) could diffuse but are immediately recaptured and trapped by the hydrogel under the membrane, resulting in $D[S_H] \leq D_{min}$.

2. Page 5: Also the DH, I requires a clear definition. Does DH, I indicate the average diffusion coefficients of DH and DI of a single TOM-CC complex at SH and SI state? Or it stands for “DH and DI”?

We thank the reviewer for pointing out the ambiguities in the definition of the diffusion coefficients of moving and non-moving TOM-CC particles in states SH and SI. To improve the definitions, we henceforth omit the designations DH, DI as well as DH,I throughout the revised version of our manuscript, as described in point 1. Thus, DH,I cannot be confused with a mean value of the diffusion coefficients $D(S_H)$ and $D(S_I)$.

In the revised version of our manuscript (page 5, lines 238–289), all diffusive TOM-CC molecules moving in the membrane and in the fully open channel state SH are described with a diffusion coefficient $D(S_H) > D_{min}$. The diffusion coefficient of non-moving TOM in SI is given by $D(S_I) \leq D_{min}$. The diffusion coefficient of non-moving molecules in SL is given by $D(S_L) \leq D_{min}$.

3. Page 5: The classification of three types of events was confusing. It would be helpful if the authors include a representative trace for each class of events.

We thank the reviewer for this useful suggestion to improve our manuscript. We have now included a new additional figure (Fig. 5a) with a representative curve for each event class from original data.

4. In the “Methods”, both *mg ml⁻¹* and *mg/ml* were used in the manuscript. Please chose one form.

All units of protein concentration are now given in the notation *mg/ml*.

5. Page 8 Line 75, unfinished sentence “labeled with the fluorescent dye Cy3 according to...42”

We apologize for overlooking to insert the name of the author of reference 42. It has now been included in the revised manuscript.

Reviewer #3:

In this manuscript, authors provide robust evidence supporting that the TOM core complex (TOM-CC), a membrane β -barrel protein, has its function impacted not just by biochemical, but also mechanical stimuli. Authors developed a simplified in vitro system to observe TOM-CC activity and diffusion, which does not fully mimic physiological contexts, but allows the identification of a correlation between TOM-CC movement and cis-trans influx. These are very interesting findings, especially because the physiological role of mitochondrial function in cell homeostasis has been extensively investigated in different cell types, strengthening a causal link between mitochondrial metabolism and cell fate and function.

We thank the reviewer for his/her insightful comments to further clarify our conclusions and improve our manuscript. Although we used a simplified in vitro system to observe TOM-CC activity and diffusion, we are glad the reviewer appreciates the importance of our study.

We have considered all recommendations point-by-point in the response below.

There are, however, some points that would need to be addressed for further clarification of authors conclusions.

1. *Page 3 Line 19: Authors raise the possibility that “high and intermediate intensity levels correspond to two conformational states”, which would reflect pore opening. This idea is sustained along the manuscript. Is there a way to address that experimentally? Or this is justified by the 3 observed bands of fluorescence intensity?*

Indeed, at present we consider the high, intermediate and low intensity levels to correspond to different conformational states of the TOM-CC. This is because the intensity levels show a simple relation to each other. The intensity values in S_H state are virtually twice those of the S_I state, which is well explained by two or one pores opening, respectively.

The different intensity levels observed in our manuscript show different levels of calcium flux through the membrane which can only be mediated by a modification of the Tom40 channel permeability. The simplest mechanistic explanation would be a conformational change of the protein. We agree with the referee that this should be studied further using complementary physical techniques. Future possibilities include the mutagenesis of both Tom40 and Tom22 at possible hinge regions within the proteins but also within the Tom40 pore. Additional approaches might be to visualize different conformations of TOM-CC using CryoEM, which are present within the molecular population. Alternatively, one might perform single molecule FRET experiments to study the dynamics of the TOM-CC between different conformations. In the Discussion section (page 7, lines 353-356) we have now indicated this.

Page 6 line 94, authors say that “Functional reversible disassembly and reassembly of the TOM-CC at time scales studied here are highly improbable.” How can authors exclude the presence of “single barrels”?

We agree that we cannot unambiguously rule out the presence of individual Tom40 barrels, although analyses of blue native PAGE and size exclusion chromatography (Ahting et al. 1999 J. Cell. Biol. 147:959-68) tend to rule out this possibility in our TOM-CC preparations. In addition, the physical probability that two single Tom40 barrels will spontaneously re-associate after dissociation rather diffuse away from each other in a 2D circular distribution is extremely small.

Nevertheless, to exclude the presence of significant single barrels of Tom40 in our experiments with TOM-CC, we purified monomeric Tom40 from *N. crassa* mitochondria and compared Tom40 with one pore (Ahting et al. 2001 J. Cell. Biol. 153:1151-1160) with TOM-CC with two pores (Ahting et al. 1999

J. Cell. Biol. 147:959-68) in terms of lateral diffusion and open-closed channel dynamics. In contrast to TOM-CC, in our many experiments, Tom40 did not show any stop-and go behavior. Tom40 followed a classical lateral Brownian motion and showed only one intensity state. TOM-CC, however, revealed different characteristics with three intensity levels (Fig.2c, Figs. S3, S9, S10 and Movies S1, S2, S3).

To clarify this point in the revised manuscript (page 3, lines 143-146) we now state:

“In contrast to TOM-CC, neither protein channel exhibits gating transitions between specific permeability states. The toggling of the TOM-CC between the three different permeability states SH, SI and SL (Fig.2c) therefore must have another root cause.”

Finally, if both pores are closed, what justifies the leakage of fluorescence observed in the low intensity level conformation?

Even in closed channels there may be a sufficient thermal fluctuation to allow stochastic leakage of calcium through the pore. We have now indicated this in our manuscript on page 3 (lines 122-124) and state:

“The low intensity level may represent a conformation (S_L) where both pores are closed with residual permeation for calcium.”

2. The use of Ni-NTA modified agarose indeed suggests there is a causal link between trapping and immobility, which leads to higher frequencies of SI and SH conformations (Fig. 4 and Fig 5). However, what could be the molecular trap or anchor point that would trigger a short-interaction between TOM-CC (Tom22) and anything below the membrane in the supporting hydrogel (first mentioned in Page 4 Line 71)?

We thank the reviewer to pick up this point. Examination of the cryo-EM structure of *N. crassa* TOM-CC (Bausewein et al. 2017, Cell 170: 693-700, Bausewein et al. 2020, Biol. Chem. 401:687-697) shows that its two Tom22 subunits protrude significantly into the IMS space, preventing direct interaction of the Tom40- β barrels with the agarose matrix. It is therefore likely that only the two Tom22 subunits are primarily responsible for the matrix-dependent channel closing activity. As recognized by the referee, this is supported by our observation that His-tagged Tom22, when tightly bound to Ni-NTA agarose, triggers permanent channel closure corresponding to state S_L . Restricting the second Tom22 by binding to Ni-NTA agarose may lead to state S_L and full channel closure. Binding of one His-tagged C-terminus of Tom22 to a Ni-NTA anchor point may cause permanent immobilization of the TOM-CC. In contrast, brief interactions could be triggered by random interaction of the C-terminus of Tom22 with the polysaccharide strands of Ni-NTA agarose near the membrane, similar to unmodified agarose.

To clarify this point, we have included an additional figure to the revised version of our manuscript showing an AFM image of a DIB-supporting agarose hydrogel (Supplementary Fig. S1). The figure shows that the thickness of the hydrogel film varies by about 40 nm. The irregular surface is consistent with Tom22-mediated intermittent binding of freely diffusing TOM-CC molecules with the polysaccharide filaments of the agarose film near the membrane. To further support permanent binding of TOM-CC to Ni-NTA modified agarose via 6xHis Tom22 (Movie S7), we now provide an additional control video (Movie S8) showing that imidazole abolishes permanent binding.

On page 5 (lines 207-209) we now state: “Based on these results, we concluded that the arrest of TOM-CC in a lipid bilayer membrane caused by short-lived interaction with the polysaccharide strands of the supporting hydrogel (Supplementary Fig. S1) triggers a transient closure of its two β -barrel pores.”

In movie S3, for example, it is clear that a SH TOM-CC moves very close to a SI TOM-CC. This could mean a very specific interaction/stopping point below the membrane is being occupied. Can authors exclude that the lipid bilayer itself or the matrigel architecture could be causing this phenomenon?

We agree with the reviewer that two single TOM-CC molecules in movie S3 are transiently trapped in almost the same location in the membrane, indicating a very specific interaction, binding or anchoring site under the membrane. A similar behavior is observed with fluorescently labeled TOM-CC crossing a specific point on the membrane a second time in movie S4.

The most plausible interpretation for this phenomenon is a transient tethering of the TOM-CC with the agarose hydrogel under the membrane. In fact, we are sure that the architecture of the hydrogel under the membrane plays an important role in determining the type of lateral membrane diffusion, which is either free Brownian motion (non-tethered) or local confinement (transient or permanent (Ni-NTA

experiment) tethered via Tom22). The latter has a strong influence on the closure probability of the two Tom40 channels within the TOM-CC.

As described above, we have included an additional figure showing an AFM image (Supplementary Fig. S1), which shows that the thickness of the hydrogel surface layer under the membrane varies considerably. This irregular surface is consistent with Tom22-mediated intermittent binding of various freely diffusing TOM-CC molecules to the same “islands” of polysaccharide filaments located in close proximity to the membrane. The fact that neighboring mobile and immobilized TOM-CCs exhibit high and medium/low intensities, respectively, suggests that the observed different intensities are not caused by the hydrogel architecture itself but by the spontaneous interaction of the TOM-CCs with the polysaccharide strands in close proximity to the membrane. In this context, it may be interesting in the future to see how Tom22-mediated immobilization of TOM by tethering to components the mitochondrial inner membrane affects the channel activity of TOM in vivo.

Since the lipids used in our bilayer do not show a phase transition at the experimental temperature and can therefore be considered as a homogeneous liquid crystalline phase, we can safely rule out an immobilizing effect due to the lipid bilayer.

Authors mention in Page 5 line 39 that “Another possible but unlikely explanation is a spatial void of agarose network”. Could authors definitely exclude that the hydrogel architecture could be having an impact by imaging it?

We believe that the AFM image of our agarose film (Supplementary Fig. S1), now included in the manuscript, confirms our original spatial concepts derived from chemical considerations.

3. Could authors explore whether any loss-of-function Tom22 mutants could have their functional impairment explained by changes in their mechano-sensitivity?

Unfortunately, there are currently no loss-of-function mutants of TOM-CC for *Neurospora* known to regulate Tom40 channel permeability. We therefore compared the behavior of TOM-CC in membranes supported by non-modified and N-NTA-modified agarose hydrogels with that of purified Tom40.

While the 6xHis TOM-CC shows an easily recognizable stop behavior, this is hardly observable with Tom40 (Movie S5 and Supplementary Fig. S4). Although the construction of permeability-loss- mutants would certainly be interesting for future studies, this requires a considerable experimental effort, which is outside the scope of the present work.

4. Signal to noise ratio in video S7 is much lower for the TOM-CC channels tracked on Ni-NTA modified agarose. Could this suggest that permeability is generally compromised by Tom-22 interaction with the hydrogel network? Was this consistently the case when using Ni-NTA modified agarose?

The reviewer is correct that the signal-to-noise ratio is lower for the TOM-CC channels traced in membranes on Ni-NTA-modified agarose. We note that the Ni-NTA agarose was custom synthesized for us to enable spin coating and is therefore not completely identical in microscopic structure to the commercially available unmodified agarose. In addition, the blue color of the modified gel may reduce the observed final intensity due to an optical filter effect.

5. I believe authors should further discuss what could be the physiological relevance and implications of TOM-CC's mechano-sensitivity. Previous reports, also referred to by the authors, show that TOM-CC can interact with different proteins in mitochondrial outer- and inner-membranes, and these different interactions lead to different functional outcomes. Based on the author's results, TOM-CC is in an open conformation when moving: how would this work in a real mitochondrion, where TOM-CC facilitates influx into the inner membrane or matrix by interacting with other proteins? Do authors suggest these interactions then happen with other proteins that are also highly diffusive?

We are aware of the fact that many cell biologists are sceptical about the physiological significance of an artificial in vitro system. For this reason, although we are convinced that the mechanosensitive effects reflect a physiological phenomenon, we have refrained from including possible physiological interpretations. However, as the reviewer has now encouraged us to do so, we have included some mechanistic speculations (clearly defined as such) in the Discussion section of the revised manuscript. This latter effects notwithstanding, the essential stop-and-go behavior of TOM-CC still shows the same type of effect observed for non-modified agarose.

On page 8, lines 346-356 we state:

“With regard to the mechanism of TOM-mediated protein translocation into mitochondria, our data open the exciting possibility that proteins at the periphery of the outer membrane, e.g. components of inner membrane TIM23 complex, not only limit the diffusion of the TOM-CC by transient interaction and formation of an active supercomplex at mitochondrial contact sites (Gomkale et al. 2021, Nat. Com. 12: 5715), but might also regulate the open-close status of the channel as a consequence. This could in turn translate into conformational changes of the TOM-CC that influence the translocation process of incoming mitochondrial preproteins. In this context, it might be interesting to visualize different Tom22-dependent conformations of TOM-CC using cryo-EM and to perform single-molecule FRET experiments to study the dynamics of TOM-CC between different conformations in the near future.”

Minor points:

1. In Fig. 1: authors might consider changing the colour of Ca²⁺ (now red and described as red in the Fig. legend) as the red is also used in the lipid membrane.

We thank the reviewer for pointing out this inconsistency. We have now corrected this and colored the lipids uniformly blue on both the left and right side.

2. Authors never mention what is a.u. in the y axis of several graphs. I assume it is arbitrary unit, but it should appear at least in the figure legend.

The meaning of the abbreviation a.u. is now given in all figure legends.

3. In video S5 there seems to be a TOM-CC complex that presents lower fluorescence intensity (compatible to S1) than the highlighted ones that is also moving.

We agree with the reviewer that video S5 shows a moving spot with low fluorescence intensity compatible with TOM-CC in S1. Since this video was taken with purified Tom40 and not from TOM-CC, we could only explain this by a residual class II TOM-CC with non-functional Tom22, as described in Fig. 5, or alternatively by a non-functional Tom40. We now address this point openly in the legend to Movie S5.

REVIEWERS' COMMENTS:

Reviewer #1 (Remarks to the Author):

In the revised version of the work by Wang et al, the authors have satisfactorily addressed all my comments. I would like to thank them for doing so in a detailed and respectful manner.

In particular, I highly appreciated the updated abstract, the new literature references, and especially the newly included OmpF control analysis (Fig. S6 and S7). This is indeed a relevant negative control.

While the diffusion-dependent pore activity of TOM-CC remains to be confirmed in vivo, this work highlights the power of in vitro measurements.

For these reasons, I support the publication of the manuscript in Communications Biology.

I wish the authors the best in their future projects.

Reviewer #2 (Remarks to the Author):

The revised manuscript and the rebuttal have addressed all issues raised in the previous review cycle. I recommend the manuscript to be published in Communications Biology.

Reviewer #3 (Remarks to the Author):

I am very impressed by the point-by-point reply prepared by the authors, which addressed every suggestion/concern raised by all 3 reviewers. The manuscript had its clarity improved, while conclusions are now also better aligned with the data. Even if still speculative, I believe it is still valid to discuss a potential physiological interpretation of the results obtained in the in vitro system developed by the authors. This reviewer recommends the manuscript in its current format for publication in Communications Biology.